# Unzipped genome assemblies of polyploid root-knot nematodes reveal unusual and clade-specific telomeric repeats

Ana Paula Zotta Mota [1] ✉, Georgios D. Koutsovoulos[1], Laetitia Perfus-Barbeoch[1], Evelin Despot-Slade [2], Karine Labadie [3], Jean-Marc Aury [4], Karine Robbe-Sermesant[1], Marc Bailly-Bechet[1], Caroline Belser [4], Arthur Péré [1], Corinne Rancurel [1], Djampa K. Kozlowski[1,5], Rahim Hassanaly-Goulamhoussen [1], Martine Da Rocha[1], Benjamin Noel [4], Nevenka Meštrović[2], Patrick Wincker [4] & Etienne G. J. Danchin [1] ✉

Using long-read sequencing, we assembled and unzipped the polyploid genomes of *Meloidogyne incognita*, *M. javanica* and *M. arenaria*, three of the most devastating plant-parasitic nematodes. We found the canonical nematode telomeric repeat to be missing in these and other *Meloidogyne* genomes. In addition, we find no evidence for the enzyme telomerase or for orthologs of *C. elegans* telomere-associated proteins, suggesting alternative lengthening of telomeres. Instead, analyzing our assembled genomes, we identify species-specific composite repeats enriched mostly at one extremity of contigs. These repeats are G-rich, oriented, and transcribed, similarly to canonical telomeric repeats. We confirm them as telomeric using fluorescent in situ hybridization. These repeats are mostly found at one single end of chromosomes in these species. The discovery of unusual and specific complex telomeric repeats opens a plethora of perspectives and highlights the evolutionary diversity of telomeres despite their central roles in senescence, aging, and chromosome integrity.

Telomeres are nucleoprotein complexes capping and protecting eukaryotic linear chromosomes. They play multiple central functions such as protection against chromosomal fusions or preventing recognition of chromosome ends by DNA damage response pathways. Telomeres also prevent the loss of genetic information during DNA replication and are involved in aging and senescence. Their malfunction can lead to severe disease including uncontrolled cell proliferation and cancers[1].

Telomeres possess an ensemble of evolutionarily conserved features considered canonical signatures. First, telomeric DNA is usually constituted by terminal short and simple G-rich repeats at chromosome ends and can fold in G-quadruplex (G4) secondary structures. Second, to counteract their shortening at each replication, these repeats are usually re-multiplied at chromosome ends by a telomerase enzyme. Telomerase is an RNA-dependent polymerase composed of a telomerase RNA template component and a telomerase reverse transcriptase catalytic subunit, which copies this template. Third, telomeric DNA usually interacts with a complex of proteins (shelterin) and telomeric RNA (TERRA) that play diverse important roles in their stability and function.

Although this system is considered canonical in eukaryotes, including in humans, some exceptions exist[2]. For instance, *Diptera*

---

[1]Institut Sophia Agrobiotech, INRAE, Université Côte d'Azur, CNRS, 400 routes des Chappes, 06903 Sophia-Antipolis, France. [2]Division of Molecular Biology, Ruđer Bošković Institute, Bijenička cesta 54, 10000 Zagreb, Croatia. [3]Genoscope, Institut François Jacob, CEA, CNRS, Univ Evry, Université Paris-Saclay, 91057 Evry, France. [4]Génomique Métabolique, Genoscope, Institut François Jacob, CEA, CNRS, Univ Evry, Université Paris-Saclay, 91057 Evry, France. [5]Université Côte d'Azur, Center of Modeling, Simulation, and Interactions, 28 Avenue Valrose, 06000 Nice, France. ✉e-mail: anazottamota@gmail.com; etienne.danchin@inrae.fr

insects seem to have ancestrally lost their telomerase[3]. Consistent with the absence of telomerase, simple telomeric repeats are not found at chromosome ends in several *Diptera* genomes. Some species display telomeric retrotransposons either instead of simple telomeric repeats or in association with them[2,4]. For example, in *Drosophila melanogaster*, three different retrotransposons, distantly related to the Jockey superfamily are specifically copied and pasted at chromosome ends by reverse transcription and play the role of telomeric DNA[5,6]. Besides, other *Diptera*, such as *Chironomus thummi* mosquitoes, possess complex repeats with no evident similarity to retrotransposons and their multiplication mechanism remains unknown[7].

In nematodes, it has been assumed that a (TTAGGC)n DNA repeat multiplied by a telomerase is the canonical system. Indeed, the repeat and the enzyme are both evolutionarily conserved between distant nematode species, including in *C. elegans*[8]. The telomerase enzyme is encoded by the *trt-1* gene in *C. elegans* but the RNA template of the repeat has not yet been identified[8]. In the animal parasite *Parascaris univalens*, a non-G-rich motif (TTGCA)n was initially described to be repeated albeit degenerated and to play the role of telomeric DNA in the somatic genome of this species[9]. However, a more recent study showed that the germline chromosomes of this species are further capped by terminal (TTAGGC)n repeats that follow the (TTGCA)n and other more degenerate repeats[10]. Selective DNA elimination removes all germline telomeric repeats and other satellite DNA in somatic cells. Fluorescent in-situ hybridization (FISH) showed that (TTAGGC)n telomere repeats are afterward re-generated at the extremities of all the somatic chromosomes. Analysis of the genome of the unichromosomal nematode *Diploscapter pachys* revealed no evidence for the canonical nematode telomeric repeats, even in the raw genome reads[11]. The study suggested that the canonical nematode repeat was also absent from the genomes of the vertebrate parasite *Trichinella spiralis* as well as the root-knot nematode *Meloidogyne hapla*. These observations suggest different repeats could protect chromosome ends in these species and the (TTAGGC)n repeat might not be as universal as assumed in nematodes.

Shelterin proteins form bridged telomeric protein complexes and are involved in many telomeric functions like telomerase regulation, protection against activation of the DNA damage response, chromosome stability, long and short-range transcriptional regulation and meiosis[12]. The shelterin complex is usually composed of both single-strand and double-strand telomeric DNA binding proteins, as well as other proteins bridging them. In humans and ciliates, there is only one POT1 (Protection of Telomeres 1), a single-strand telomere-binding protein with several oligosaccharide/oligonucleotide binding folds (OB folds). OB folds found across distant organisms commonly bind single-stranded DNA, although a wider specificity, including to other ligands was shown in some cases. The conserved OB-fold of POT1 is necessary for single-stranded DNA binding specificity and the telomere protective functions. In *C. elegans*, there are 4 distinct POT1 homologs containing only one OB fold: POT-1, POT-2, POT-3 and MRT-1[8]. POT-1 and POT-2 exhibit structural similarity to the first and second OB fold, respectively and can promote T-loop formation in vitro[13], and repress telomerase activity in vivo in *C. elegans*[14]. POT-3 exhibits structural similarity to the second OB fold and was recently found to preferentially bind the telomeric DNA repeats on the G-rich 3' overhang[15], while its absence leads to uncontrolled telomere lengthening. MRT-1, the fourth POT-1 homolog, possesses an SNM1 nuclease domain in addition to an N-terminal domain similar to the second OB-fold of POT1, and this gene is required for telomerase activity[16]. The number of POT1 homologs in nematodes seems to vary[8]. TRF (Telomeric Repeat Factors) proteins and 3'-overhang-binding heterodimers are linked by a protein bridge, formed by Rap1 and Poz1 in the yeast *Schizosaccharomyces pombe* and by TIN2 (an ortholog of Poz1) in mammals. However, in *C. elegans*, TRF protein homologs with a TRFH (Telomeric Repeat Factors Homology) domain or a telobox at their

C-terminus could not be found. Instead, in *C. elegans*, the shelterin complex contains two other proteins, TEBP1/DTN-1 and TEBP2/DTN-2, which bind double-stranded telomeric DNA and POT-1[17]. Sterility is observed after one or several generations in tebp-1 plus tebp-2 double mutants[17,18]. A single TEBP homolog was found in many nematodes, including in *C. briggsae*[18].

Although the presence of a telomerase, a simple (TTAGGC)n repeat, and a shelterin complex is considered the default telomeric system in nematodes, no extensive search has been performed in this phylum. Many nematode genomes are still highly fragmented with long repetitive regions left unassembled, which prevents identification of DNA repeats at chromosome ends. Long-read sequencing technology, elected 2022 method of the year[19] opened new perspectives toward resolving the genomes of non-model species and exploring how they start and end.

Here, using long-read sequencing, we assembled the genomes of the root-knot nematodes *Meloidogyne incognita*, *M. javanica*, and *M. arenaria*, three agricultural pests of major concern worldwide. Because these genomes have a complex polyploid structure, previous efforts yielded fragmentary assemblies[20,21]. Our long-read assemblies were two orders of magnitude more contiguous and allowed assembling the three (*M. incognita*) and four (*M. javanica*, *M. arenaria*) subgenomes separately (hereafter referred to as unzipping). We used these more accurate representations of the genomes to investigate how chromosomes start and end and whether the telomere system considered canonical in nematodes was conserved. We found that the canonical nematode telomeric repeat as well as the telomerase and other known telomere-maintenance proteins were lost in *Meloidogyne* but also multiple times independently in other nematode genera. In the three *Meloidogyne* species we have sequenced, we discovered peculiar complex repeats enriched at contigs ends and specific to these species. Using FISH, we confirmed telomeric localization of the repeats mostly at one end of the chromosomes.

## Results

### Long-read sequencing improves genome assemblies and confirms polyploidy

We assembled the *M. incognita* (*Minc*), *M. javanica* (*Mjav*), and *M. arenaria* (*Mare*) genomes in 291, 364, and 377 contigs, respectively. This represents a massive improvement compared to previous assemblies in ca. 12,000–31,000 scaffolds[20] (Table 1). The assembly sizes reached 199.4, 297.8, and 304.3 Mb for *Minc*, *Mjav*, and *Mare*, consistent with previous flow cytometry estimates[20] (Table 1). For the three species, the contig N50 lengths were around 2 Mb which is two orders of magnitude higher than N50 values for previous assemblies (0.01–0.04 Mb). These three *Meloidogyne* species are described as polyploids resulting from complex hybridization events[20–22]. Consistent with these findings, analysis of k-mer distribution in the sequencing reads (Supplementary Fig. 1) suggested a triploid (AAB) genome for *Minc* and tetraploid (AABB) genomes for *Mjav* and *Mare*, with respectively 6.6, 8.7, and 9.2% average nucleotide divergence between the homoeologous subgenome copies (Supplementary Fig. 2). Assessment of completeness via CEGMA[23] and BUSCO[24] genes shows a moderate improvement and suggests previous assemblies, albeit much more fragmented, already provided comprehensive representations of the gene contents (Table 1). Consistent with k-mer estimations of ploidy, CEGMA genes were present on average in 2.7 copies in *Minc* and 3.8 copies in *Mjav* and *Mare*. Supporting genome completeness, almost all k-mers found in the reads were present in the assembly for the three species (Supplementary Fig. 3). Finally, analysis of contigs coverage, GC content, and taxonomic assignment, revealed no contamination and allowed isolating contigs corresponding to the mitochondrial genome in each species (Supplementary Fig. 4).

To further characterize the genome structures, we used the predicted genes from EuGene[25] (Supplementary Table 1) as anchors to

**Table 1 | Genome assembly metrics**

| | Minc[a] | Minc[b] | Mjav[a] | Mjav[b] | Mare[a] | Mare[b] |
|---|---|---|---|---|---|---|
| Flow cytometry[a] | 189 +/− 15 | 189 +/− 15 | 297 +/− 27 | 297 +/− 27 | 304 +/− 9 | 304 +/− 9 |
| Assembly size (Mb) | 183.5 | 199.4 | 235.8 | 297.8 | 258.1 | 304.3 |
| N50 (Mb) | 0.04 | 1.86 | 0.01 | 2.07 | 0.02 | 2.04 |
| # Contigs | 12,091 | 291 | 31,341 | 364 | 26,196 | 377 |
| CEGMA (N: 248) | CC: 94.76 A: 2.93 | CC: 94.76 A: 2.69 | CC: 92.74 A: 3.68 | CC: 95.56 A: 3.75 | CC: 93.95 A: 3.57 | CC: 95.97 A: 3.78 |
| BUSCO metazoan (N: 954) | BC:56.3 (D:45.0) F:9.3 M:34.4 | BC:56.6 (D:47.6) F:9.1 M:34.3 | BC:56.8 (D:45.0) F:8.9 M:34.3 | BC:57.1 (D:51.5) F:8.9 M:34.0 | BC:56.6 (D:47.1) F:9.8 M:33.7 | BC:57.9 (D:51.8) F:8.6 M:33.5 |
| GC% | 29.75 | 30.11 | 29.96 | 30.20 | 29.97 | 30.29 |

CC: % of complete CEGMA genes, A: average number of orthologs per CEGMA gene. BC: % of complete BUSCO genes, D: % of duplicated complete BUSCO genes, F: % of fragmented BUSCO genes, M: % of missing BUSCO genes.
[a]Previous versions[20].
[b]This study.

**Table 2 | Classification of duplicated genes in the three *Meloidogyne* genomes**

| Feature/species | Minc | Mjav | Mare |
|---|---|---|---|
| Protein-coding genes | 52,182 | 61,060 | 59,500 |
| Singleton | 7104 (13.6%) | 5676 (9.3%) | 5294 (8.9%) |
| Dispersed dup. | 7854 (15.1%) | 7760 (12.7%) | 7507 (12.6%) |
| Proximal dup. | 1711 (3.3%) | 1277 (2.09%) | 1195 (2.0%) |
| Tandem dup. | 957 (1.8%) | 767 (1.26%) | 836 (1.41%) |
| WGD/Segmental dup. [Dup. depth peak] | 34,556 (66.2%) [3X - 54%] | 45,580 (74.65%) [4X - 65%] | 44,668 (75.1%) [4X - 69%] |
| Aligned blocks | 730 | 1,041 | 891 |
| Contigs involved | 193 (66.3%) | 246 (67.6%) | 232 (61.5%) |

detect duplications with McScanX[26] and to differentiate the sub-genome types (A or B) among contigs. Most of the genes (66−75%) were classified in the category 'whole genome duplication (WGD)/ segmental duplication', consistent with the three genomes being polyploid (Table 2). Similar analyses on previous assemblies only allowed assigning 28.5, 4.6, and 3.4% of the genes to the WGD category, in *Minc*, *Mjav*, and *Mare*, respectively[20], because of the high fragmentation and low N50 of previous assemblies. Within this category of duplications, we observed a peak of genes in blocks duplicated at a depth of 3X for *Minc* (54% of WGD) while for *Mjav* and *Mare*, we observed a peak at a depth of 4X with respectively 65% and 69% of the genes present in the WGD category (Table 2). These results are consistent with *Minc* being triploid while *Mjav* and *Mare* being tetraploid. For all the species, more than 60% of the contigs form duplicated blocks with other contigs.

To assign the subgenomes of *Minc*, we took into account all the triplicated blocks encompassing at least 10 collinear genes. We found 116 blocks with these characteristics comprising 16,892 genes in 159 contigs. Using the same strategy extended to tetraploidy, we could identify 113 synteny blocks for *Mare* and 145 for *Mjav*, comprising 21,521 and 21,456 genes, respectively.

We computed the Ks (rate of synonymous mutations) values for each gene pair within the blocks, and for each contig pair we calculated the median Ks value. Consistent with an AAB genome, we observed a two-peak distribution in *Minc*, with a relatively lower Ks value representing the A-A divergence and a higher value representing the divergence between A and B (Supplementary Fig. 5). Using this property on the triplicated contigs, we could assign unequivocally 158 contigs to either one of the two A-A' sub-genomes (113 contigs) or the B sub-genome (43 contigs). Extending the same strategy as for *Minc* but adapted to quadruplets of copies rather than triplets, we also observed a two-peak distribution of Ks values in *Mjav* and *Mare* (Supplementary

Fig. 5). Here also, higher Ks values represent divergence between As and Bs subgenomes while lower values represent divergence either between A-A' or B-B'. Because the median Ks relations between As and Bs sub-genomes in these AABB genomes are symmetrical, Ks information alone is not sufficient to determine which contigs are A or B in these species.

Therefore, to further characterize and assign A and B subgenomes to *Mjav* and *Mare* contigs, we compared collinear regions conserved between *Minc* and *Mjav* and *Minc* and *Mare* and present in 3 copies in *Minc* while in 4 copies in *Mjav* and *Mare*, using cross-species McScanX analyses. We identified 148 such conserved collinear blocks for the *Minc/Mare* comparison, comprising 19,939 genes, and 154 blocks for the *Minc/Mjav* comparison covering 18,059 genes. We constructed classification trees of the contigs based on the median Ks values between all the relationships and assigned an A or B sub-genome to *Mjav* or *Mare* contigs according to their position in the tree relative to previously assigned A and B contigs of *Minc*. In *Mare*, we could unequivocally assign 64 contigs to A-sub-genomes and 56 to B-sub-genomes, while for *Mjav* we assigned 58 contigs to A-sub-genomes and 62 to B-sub-genomes.

Finally, to investigate the relationship between the A and B sub-genomes of the 3 species altogether, we conducted a 3-species McScanX analysis and classification tree according to median Ks between cross-species conserved regions. The most frequently observed topology showed a clear distinction between all the A sub-genomes on one side and all the B sub-genomes on the other side, consistent with these species being polyploid hybrids sharing common ancestry (Supplementary Fig. 5).

Overall, the new assemblies produced much more contiguous genomes with N50 values two orders of magnitude higher than before, allowing better resolution of the duplicated genome structure and assignment of A and B sub-genomes for each species. With half of the contigs reaching several megabases for all the species, and the biggest probably nearly representing whole chromosomes, we investigated how chromosomes start and end in these species.

## *Meloidogyne* genomes lack canonical nematode telomere repeats and most telomere-associated proteins known in *C. elegans*

The chromosomes of *C. elegans* and most nematodes start and end with an evolutionarily conserved (TTAGGC)n telomeric repeat[8]. We did not find this telomeric sequence repeated at contig ends in the *Minc*, *Mjav*, and *Mare* genomes we have assembled. Furthermore, this telomeric sequence was also not found repeated at contig or scaffold ends in any of the publicly available *Meloidogyne* genome assemblies. To better investigate how evolutionarily conserved and thus how canonical the *C. elegans* telomeric repeat is in the phylum *Nematoda*, we searched in all the available nematode genomes assembled with a

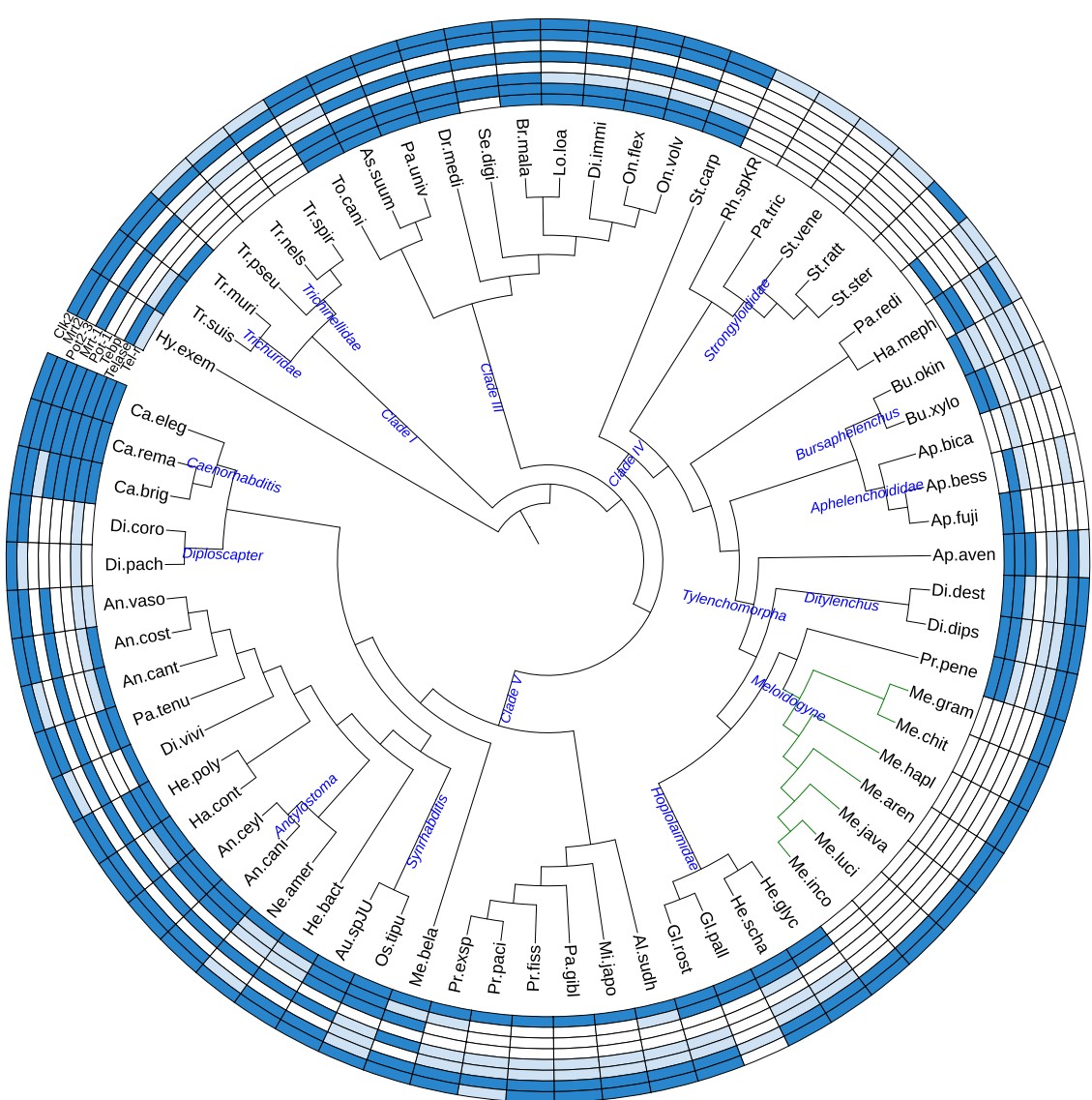

**Fig. 1 | Distribution of telomere repeats and telomere-associated proteins in the phylum *Nematoda*.** Blue color means present, light-blue means partial evidence and blank no evidence. Source data is provided as a Source Data file. Tel-r: *C. elegans* telomeric repeat (TTAGGC)n except for *Hypsibus exemplaris* in which another simple 9-nucleotides repeat has been described[64]. Telase: telomerase reverse transcriptase. Tebp: ds-telomeric DNA binding proteins Tebp1 or Tebp2. Pot1: ss-telomeric DNA binding protein Pot-1. Mrt1: ss-telomeric DNA binding protein Mrt-1. Pot2-3: ss-telomeric DNA binding proteins Pot2 or Pot3. Mrt2 and Clk2: proteins with a putative function in telomere length regulation. In green lines, *Meloidogyne* genus.

minimal N50 length of 80 kb. In contrast to the *Meloidogyne* genus, (TTAGGC)n could be identified repeated in arrays in the genome assemblies for most other nematode genera investigated, including in *Pratylenchus penetrans*, the most closely related species with a genome available (Fig. 1, Supplementary Data 1). In the rest of nematode clade IV, to which the *Meloidogyne* belongs, the canonical telomeric repeat was well conserved except in the *Strongyloididae* group of animal parasites. Overall, ancestral state reconstruction using parsimony, suggests the *C. elegans* telomere repeat was present in the last common ancestor of nematodes but lost in *Trichinellidae* (clade I), in *Diploscapter* (clade V) and among Clade IV in the *Strongyloididae* and *Meloidogyne* groups (Supplementary Fig. 6). Further search for arrays of perfectly conserved nucleotide repeats of size 6 to 12 at contig ends of *Minc*, *Mjav*, and *Mare* also failed to identify any other simple telomere repeats in these species.

Because no evidence for the canonical nematode telomeric repeat or other simple repeat arrays were found at contig ends, we investigated whether a telomerase enzyme was present in *Meloidogyne* and other nematode genomes. Combined search of protein motifs, and homology search in the nematode genomes revealed no evidence for a telomerase enzyme in any of the *Meloidogyne* genomes investigated, not even at the pseudogene level. This observation is consistent with the absence of the canonical *Nematoda* telomeric repeat in the *Meloidogyne*. Within clade IV, besides the *Meloidogyne*, evidence for telomerase enzymes was found in most other nematodes, including in *P. penetrans*, the closest outgroup species relative to the *Meloidogyne* (Fig. 1, Supplementary Data 1). However, no evidence for a telomerase enzyme could be found in the *Strongyloididae*, which also showed no evidence for the (TTAGGC)n telomeric repeat. Extending the analysis to all the other nematode clades allowed the identification of telomerase enzymes in all species in clade III. However, in clade V, although the *Caenorhabditis*, *Diploscapter* and *Ancylostoma* genera showed evidence for a telomerase, the rest of the genera either displayed sparse evidence or no evidence for a telomerase. Finally, in clade I, no species presented strong evidence supporting the presence of a telomerase enzyme. Ancestral state reconstruction suggested a telomerase enzyme was present in the

nematode ancestor but secondarily lost multiple times in the *Meloidogyne*, *Strongyloididae*, *Trichinellidae*, and *Diplogastridae* groups (Supplementary Fig. 7). The absence of detectable telomerase suggests an alternative lengthening of telomere (ALT) pathway is present in *Meloidogyne* and several other genera of nematodes.

Finally, we investigated whether shelterin complex and other telomere-related proteins from *C. elegans* were conserved. No evidence for homologs of single-stranded telomeric repeats-binding proteins POT-1, 2, or 3, or MRT-1, or even their characteristic domains, could be identified in any *Meloidogyne* species (Fig. 1, Supplementary Data 1). In the rest of clade IV nematodes, some species showed evidence for an ssDNA-binding domain of telomere protection protein, including *P. penetrans* as well as other groups of plant-parasitic nematodes such as *Hoplolaimidae* (*Globodera* and *Heterodera*), *Ditylenchus* or *Bursaphelenchus*. However, these proteins had no recognizable orthology relationships with those of *C. elegans*. This suggests orthologs of *C. elegans* proteins have highly diverged or other proteins might have been recruited to bind telomeric DNA in these nematodes. In clade I and clade III nematodes, single-stranded telomeric DNA-binding domains were found but orthology evidence could be identified only for MRT-1, suggesting the rest of POT-1 homologs were absent. Finally, in clade V, the four POT-1 homologs could be identified in *C. elegans* and *C. remanei* but only two of them (MRT-1 and POT-1) in *C. briggsae*. In the rest of clade V, evidence for a ssDNA-binding domain of telomere protection proteins could be identified in a majority of species. Yet, orthology relationships could be confirmed only for MRT-1 in a few species and several species showed no evidence for the presence of any of the four POT-1 homologs (e.g. *Diploscapter* genus). This ensemble of observations supports the idea that the number of POT-1 homologs varies from 0 to multiple copies in nematodes.

Similarly to single-strand telomere-binding proteins (i.e. POT-1 homologs), no evidence for homologs of *C. elegans* double-strand telomeric DNA-binding protein (TEBP) or domain could be found in any of the *Meloidogyne* genomes (Fig. 1, Supplementary Data 1). More broadly, apart from *Aphelenchus avenae* and *Halicephalobus mephisto* no evidence for a dsDNA-binding domain of telomere-associated protein could be identified in the rest of clade IV or any species from clade I. In contrast, most nematode species from clade III showed evidence for homologs of these proteins. Finally, in clade V, apart from the *Caenorhabditis*, the *Synrhabditis*, and *Ancylostoma* groups, in which TEBP homologs could be identified, most other genera also showed either no or sparse evidence for the conservation of these proteins or associated domains.

Besides single and double-stranded telomeric DNA-binding proteins which can be considered equivalent to the shelterin complex in *C. elegans*, other proteins are described as playing a role in telomere maintenance or elongation. MRT-2 is a homolog of yeast RAD17 and human RAD1 DNA damage checkpoint proteins, and *C. elegans* mutants *of mrt-2* show defects in germline immortality associated with telomere shortening[27]. It has been proposed that MRT-2 regulates telomere length but does not necessarily directly bind telomeric DNA[18]. Our study shows MRT-2 is widely conserved in nematodes, including in the *Meloidogyne* species, as well as all clade I and clade III species. This is supported by both the presence of the protein domain and orthology relationship (Fig. 1, Supplementary Data 1). Within clade IV, MRT-2 protein domains seem to be absent from the plant-parasitic *Aphelenchoididae* as well as from *Strongyloididae* animal parasites. In clade V, MRT-2 was conserved in 20 of the 25 species, covering all nematode superfamilies in this clade. Overall, the strong conservation of MRT-2, even in species that display neither canonical repeats nor telomerase enzymes, suggests that besides the putative role in telomere length regulation, this protein plays a core central role in nematode DNA damage checkpoint.

CLK-2/TEL-2 also known as TELO2 in humans is a telomere maintenance protein essential for viability and widely conserved across eukaryotic evolution from yeasts to vertebrates[28]. Although a role in telomere length regulation has initially been proposed[29,30], the exact function of CLK-2 relative to telomeres remains unclear because mutants display pleiotropic effects and it was also suggested that CLK-2 is mainly a DNA damage checkpoint protein[28,31]. We found CLK-2 widely conserved in nematodes with some exceptions. In clade I, although the PANTHER[32] domain PTHR15830 'Telomere length regulation protein Tel2 family member' is present in all the species, no orthology relationship with *C. elegans* CLK-2 could be identified in the *Trichinellidae*. In contrast, in clade III, both the protein domain and orthology relationship were identified in all the species included in our analysis. In clade IV, both the domain and orthology relationship could be identified in the *Meloidogyne* species but Clk-2 appears to be absent from the whole *Aphelenchoididae*, and only sparse evidence was found in *Strongyloididae*, similarly to MRT-2. In clade V, candidate CLK-2 orthologs could be identified in all the species included in our analysis except *Pristionchus pacificus* although the protein domain was present.

## Complex candidate telomeric repeats are enriched at some contig extremities

All the *Meloidogyne* genomes available to date lack the canonical telomeric repeats of nematodes and no other simple repeats were found at contig ends in our assemblies. Furthermore, no orthologs of telomerase or any of the single- or double-stranded telomeric DNA-binding proteins known in *C. elegans* were found in these species. In the absence of telomerase, ALT-mediated telomere lengthening is associated with notable variations in telomeric sequences[33]. Therefore, we searched for enriched motifs at the contigs extremities of *Meloidogyne* genomes, allowing imperfectly repeated sequences. In each species, we identified three enriched repeated motifs ranging from 80 to 100 bp (Fig. 2, Supplementary Figs. 8–10). Comparison of all the motifs revealed one motif (motif-1) was highly conserved within and between species and formed a first clade (Fig. 2, square symbol). The two other motifs (motif-2 and motif-3) of each species showed more variations. The second clade includes three motifs that share a completely conserved 27-bp region. Among them, *Mjav* motif-2 (MjM2) and *Mare* motif-3 (MaM3) were highly similar sequences while *Mare* motif-2 (MaM2) was more diverged (Fig. 2 - circle symbol). All the motifs in this clade were relatively G-rich. Finally, a third clade consists of three divergent motifs, two from *Minc* (MiM3 and MiM2) and one from *Mjav* (MjM3). They were characterized by stretches of Gs interspersed by more variable regions (Fig. 2, star symbol). The whole repeat unit was composed of the three motifs albeit with a different relative order between species (Fig. 2). For each species, retrieving and aligning sequences of all the repeat units enriched at the contig extremities allowed deducing ca. 250–300 bp consensus sequences (Supplementary Figs. 11–13).

In the *Minc* genome, the consensus sequence was present in 58 contigs and formed repeat arrays on 57 of them (Fig. 3, Supplementary Fig. 14, Supplementary Data 2). Repeat arrays contained 8 to 187 units and all comprised at least motifs 1 and 2 while 49 contained the three motifs. These repeat arrays spanned regions ranging from 2.3 to 59.3 kb and were present exclusively either at the beginning or at the end for 51 of the 57 contigs (~88%). For one large contig (#10) the arrays started 35 kb away from the beginning and 5 shorter contigs ranging in length from 2.4 to 43.8 kb were entirely made of these repeat arrays. Thus, excluding those contigs, the repeat arrays were present at the beginning of 25 contigs and the end of 26 other contigs but never present at both extremities. We noted that, when present at the beginning of contigs, the consensus sequence was repeated on the positive strand while when present at the end, it was repeated on the negative strand in reverse complement.

The *Mjav* composite repeat formed 59 arrays in 57 contigs ranging from 3 to 22 units (Supplementary Fig. 15, Supplementary Data 3). No contig was entirely repetitive. In 30 contigs, the repeat was exclusively at the beginning and systematically on the positive strand. In 19

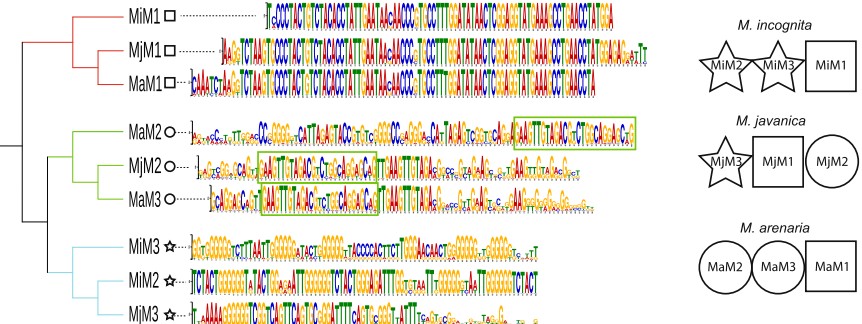

**Fig. 2 | Cross-species comparison of motifs found enriched at the contigs extremities of *Minc, Mare,* and *Mjav*.** Cladogram based on the alignment of the motifs enriched at the contig extremities in the three sequenced species. Each color in the logo represents a different nucleotide, and their size is proportional to the conservation across all the matches. The symbols at the end of each branch indicate the motif type. The green box indicates the similar sequence among the motifs MaM2, MjM2 and MaM3 of *Mare* and *Mjav*. On the right side, the suite of motifs characteristic for each species repeat unit.

contigs, the repeat was exclusively at the end and systematically on the negative strand. In 6 contigs, the repeat was several kilobases away from either the beginning or the end and with interspersed positive and negative orientations. Finally, we identified two contigs (6 and 11) where the unit was repeated both at the beginning and at the end, respectively exclusively in the positive and negative strands.

In *Mare*, we identified 57 repeat arrays ranging from 3 to 127 units in 53 contigs plus one contig being fully repetitive (Supplementary Fig. 16, Supplementary Data 4). In 44 contigs the unit was repeated either exclusively at the beginning or the end and respectively exclusively in the positive strand or negative strand in 20 and 19 of these contigs. In 7 other contigs, the unit was repeated either at several kbs from the beginning only (5) or from the end only (2) with interspersed orientations. Finally, in one contig we observed repeat arrays at several kilobases of both the beginning and the end, and in one other, we found an array at the beginning and another at >1 Mb from the end. In these two cases (contigs 1 & 2) interspersed positive and negative orientations were observed near the beginnings and the ends.

In the three species, the composite repeats were mostly repeated at one single end of the contigs. The number of repeat arrays identified is close to the observed number of chromosomes in each species, suggesting the repeats might be at one single end of most chromosomes. As could be expected, these regions were repeat-rich and gene-poor (Supplementary Data 5). We used this property to check whether other motifs were enriched in other gene-poor repeat-rich contig extremities. Using the same search strategy, allowing degenerate motifs, we did not find another repeat forming a number of arrays close to the number of chromosomes. Instead, we found multiple different repeats, each specific to a small number of contig extremities.

### The candidate telomeric repeats are different in each species and not sub-genome-specific

Our conserved synteny study confirmed the genomes of *Minc, Mjav,* and *Mare* are polyploid with A and B sub-genomes and more similarity between species within a sub-genome than within a species between A and B sub-genomes. We investigated whether the candidate telomeric repeats followed the same evolutionary history as that of the sub-genomes. Although the repeat units and relative orders of the motifs are different between species, motif-1 is highly conserved within and between species. Therefore, we compared motif-1 occurrences within and between species in the light of their assignment to A and B sub-genomes. We found 2,660, 606, and 1,743 motif-1 occurrences in *Minc, Mjav,* and *Mare* genomes, respectively. Grouping motif-1 according to sequence similarity resulted in 52 clusters (35 with multiple occurrences of motif-1 and 17 singletons). Analysis of the clusters (Supplementary Data 6) revealed that 85.5, 86.2 and 85.5% of motif-1

occurrences, in *Minc, Mare,* and *Mjav*, respectively, fell in clusters composed of both A and B contigs, indicating motif-1 does not cluster by sub-genome within species but is mostly indistinguishable between A and B sub-genomes. Extending the analysis to cross-species comparisons, we found that 4,288 motif-1 occurrences (88.5%) were in 14 multi-species clusters while only 771 were in 38 small species-specific clusters. Nine of these 14 multi-species clusters, covering 97.8% (4,192/4,288) of motif-1 occurrences, were composed of motifs coming from both A and B contigs, indicating again no separation between motifs of A and B sub-genomes.

Species-specific repeat units and clustering of motif-1 without grouping according to A or B contigs suggests that the candidate telomeric repeats have evolved independently in each species after the hybridization events rather than being inherited from putative ancestors of the A and B sub-genomes.

### The terminal repeats are specific to polyploid mitotic parthenogenetic root-knot nematodes

We investigated whether the composite repeats found in *Minc, Mjav* and *Mare* as well as the motifs that constitute them matched other nematode genomes. We searched in publicly accessible nematode genome assemblies, including those of *M. hapla, M. chitwoodi, M. exigua, M. luci,* and *M. graminicola*. We found that the *Minc* composite repeat was conserved (including the 3 constitutive motifs) in the *M. luci* genome[34]. Like in the other species, it was mostly present repeated at either contig beginnings in the positive strand or at contig ends in the negative strand, and the repeat unit was constituted of motif-2, motif-3, and motif-1, hence the same order as in *Minc* (Supplementary Data 7). This observation is consistent with a recent phylogenomic analysis of the *Nematoda* phylum suggesting a phylogenetic position of *M. luci* closer to *M. incognita* than to the other polyploid mitotic parthenogenetic root-knot nematodes[35]. However, none of the composite repeats or constitutive motifs found in the polyploid mitotic parthenogenetic root-knot nematodes were conserved in the other *Meloidogyne* species analyzed. The composite repeats and their constitutive motifs were also absent from the rest of the nematode genomes investigated, suggesting they are specific to the mitotic parthenogenetic *Meloidogyne* species. More broadly, a search against the NCBI's nt library with the *Minc* repeat as query returned one single significant hit. Interestingly, this hit was against an *M. incognita* sequence (accession # S68778.1, https://www.ncbi.nlm.nih.gov/nuccore/S68778.1/) described, as early as 1991, as a species-specific sequence useful as a diagnostic tool[36]. The match was with a region of the repeat that contains part of motif-3 and the whole motif-1. As could be expected, consensus repeated sequences of *Mare* and *Mjav* also returned this sole hit against the NCBI's nt library and only on a region corresponding to the non-degenerated motif-1.

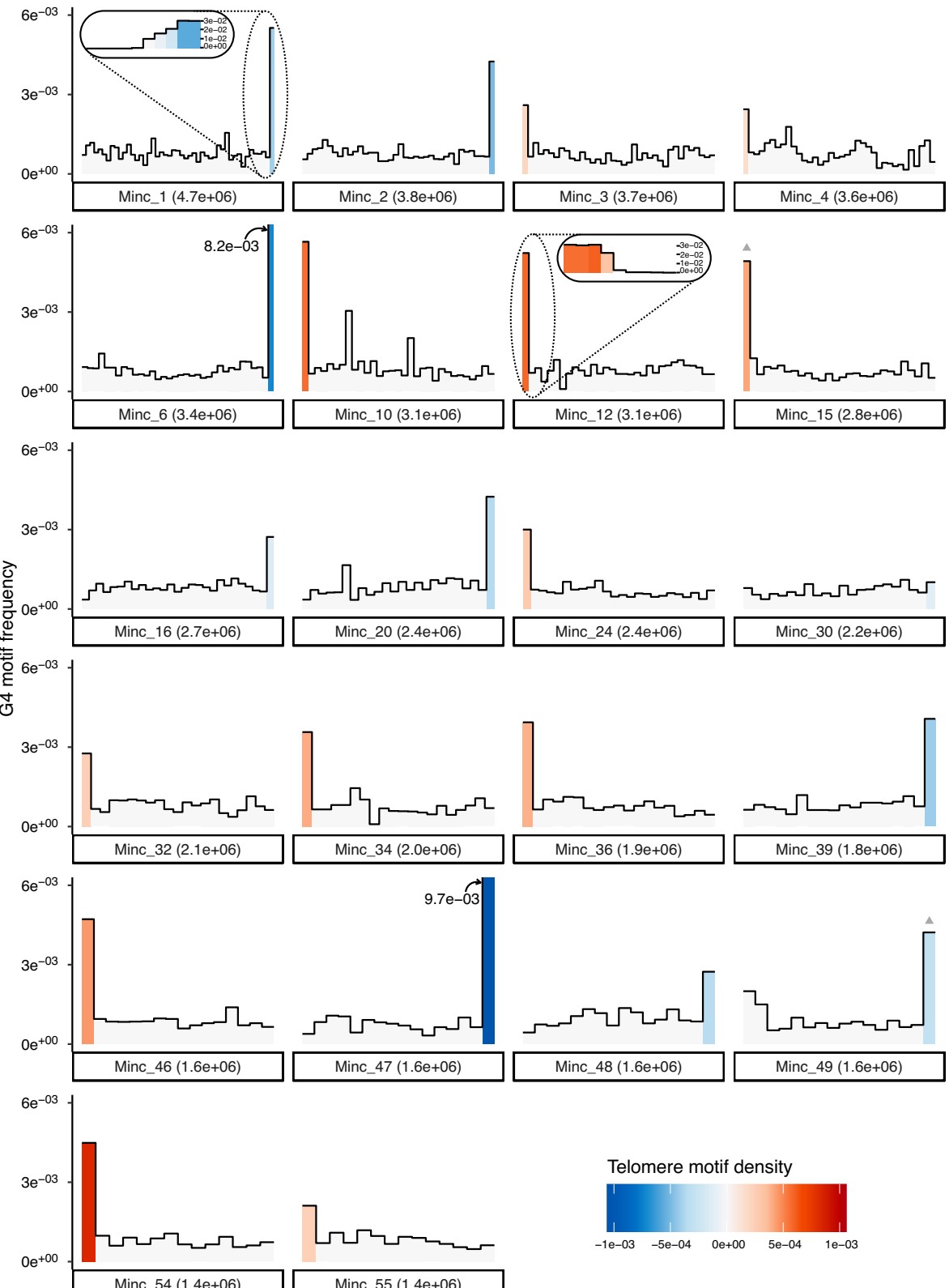

**Fig. 3 | Density of *Minc* composite repeats and G-quadruplex along genome contigs.** The density of *Minc* composite repeats per 100 kb window is represented by a color gradient with positive values (red) indicating a density in the sense strand and negative values (blue) indicating a density in the reverse complement strand. Gray triangles above bars indicate regions of the contigs where less than 100% of the repeats are on the same strand. The heights of bars in the histogram represent the density of G-quadruplex forming regions in the same windows. Only contigs containing repeat arrays and bigger than the N80 value are represented. Zoom-in representations of the 50 last kb of contig 1 and 50 first kb of contig 12 illustrate the gradual increase in repeat and G4 density in 5 kb windows towards the contig extremities. The same information for all the repeat-containing contigs is available in Supplementary Fig. 14. Source data to produce this figure is provided as a Source Data file.

Overall, apart from the clade constituted by polyploid and mitotic parthenogenetic root-knot nematodes, the composite repeats identified at contig extremities in *Minc*, *Mare*, and *Mjav* are not conserved in any other nematode genomes, or any other species represented in the NCBI's nt library so far.

## No evidence that the terminal repeats of *Meloidogyne* contigs are transposable elements

In *Drosophila* and other *Diptera* insects that lack telomerase, retrotransposons have replaced simple telomeric repeats. Furthermore, previous analyses have suggested transposable elements are active in *Minc*[37]. Therefore, we investigated whether the composite terminal repeats identified in mitotic parthenogenetic *Meloidogyne* genomes could be related to transposable elements. We used EDTA[38] to predict and annotate transposable and other repetitive elements in the *Minc*, *Mjav*, and *Mare* genomes (Supplementary Table 2). In *Minc*, EDTA predicted two repetitive sequences in the regions corresponding to candidate telomeric arrays. One repeat was unclassified and the second one was annotated as a putative CACTA TIR transposon (MITE/DTC code). Motifs 3 and 2 of the *Minc* composite repeat were present in the unclassified EDTA sequence TE00000264, while motif 1 matched the repetitive sequence annotated as a CACTA TIR DNA transposon (TE00000823). However, no evidence for a transposase or any of the protein-coding genes usually found in CACTA DNA transposon could be identified in this predicted repeat. Furthermore, searching this annotated repeat (TE00000823) against the Repbase28 database[39] returned no significant hit. Hence, this assignment as a CACTA DNA transposon is most likely an annotation error from EDTA. The consensus sequences of *Minc*, *Mjav*, and *Mare* repeat units themselves also returned no hit against Repbase28 and returned no similarity with a (retro)-transposon-related gene, further illustrating the lack of evidence for similarity to a known transposable element.

## The candidate telomeric repeats are predicted to form G-quadruplex structures in *Meloidogyne* genomes

One characteristic feature of telomeric repeats is the presence of a G/C bias with a G-rich repeat oriented towards the 3' end[40]. The G-rich telomeric regions are capable of forming four-stranded G-quadruplexes (G4) and these secondary structures are assumed to play important roles in the functions of telomeres[40]. The evolutionary conservation of G-richness in telomeric repeats was hypothesized to be related to the importance of forming G-quadruplexes. These G-quadruplexes are also suspected to play a role in the ALT telomere lengthening pathway. Motifs 2 and 3 of the three *Meloidogyne* species are relatively rich in G, with some of them (MiM3, MiM2, MjM3) even displaying consecutive stretches of Gs (Fig. 2). Therefore, we investigated whether the composite repeats of the three *Meloidogyne* species could form G-quadruplexes. Regions forming possible G-quadruplexes were predicted on the three *Minc*, *Mjav*, and *Mare* composite repeats. The *Minc* repeat returned the highest number of predicted G4 structures (103) and this concerned both G-rich motifs 2 and 3. In *Mare* and *Mjav* composite repeats, respectively 11 and 4 regions were predicted to form potential G-quadruplexes. To investigate whether G4-forming regions were a characteristic feature of our candidate telomeric repeats, we also predicted these secondary structures on the whole genomes. In *Minc*, we could observe a clear enrichment of segments forming G-quadruplexes in the genomic regions corresponding to the terminal repeat arrays, with the rest of the genome being otherwise poor in G4-forming segments (Fig. 3, Supplementary Fig. 14). Similar distributions were observed in the *Mjav* and *Mare* genomes (Supplementary Figs. 15 and 16) albeit less marked than in *Minc*.

## The *Meloidogyne* candidate telomeric repeats are transcribed

TElomeric Repeat-containing RNA (TERRA) results from the transcription of telomeric regions whether they are constituted of simple repeats like in most eukaryotes, or more complex repeats like transposons in *Drosophila melanogaster*[4,41]. These transcribed telomeric regions constitute a common feature of eukaryotic genomes and they are suspected to play important regulatory roles, including in humans[1,42]. For instance, TERRA in mammals is involved in the formation of RNA−DNA hybrid structures, known as R-loops, which play a role in telomere maintenance in ALT-positive cells. We analyzed de novo assemblies of short-read RNA-seq transcriptome data previously published for the same *Minc* isolate[20] for four different developmental life stages: eggs, pre-parasitic second-stage juveniles (J2), a mixture of late J2, J3 and J4 parasitic stages (J3-J4) and adult females. We identified at least one of the three constitutive motifs of the *Minc* composite repeat in 65 predicted isoforms from 14 assembled transcripts, suggesting the telomeric repeats are transcribed (Table 3, Supplementary Data 8). Two isoforms of a transcript in the J2 stage transcriptome contained both MiM1 and MiM2. In the egg stage, one transcript had two isoforms containing both MiM2 and MiM3 while a third isoform of the same transcript contained MiM1. The rest of the isoforms contained only one of the three motifs. Difficulty in assembling sequences containing repetitive low-complexity regions de novo from short-read data might explain why no single isoform containing the three motifs together was observed. For 9 out of the 14 transcripts containing the repeat motifs, at least one isoform mapped to the *Minc* genome assembly with at least 94% identity on at least 80% of their length. Three other transcripts had partially mapped isoforms and two other transcripts had no isoform mapping the genome (Supplementary Data 8). As short-read transcriptome data could be prone to assembly errors on highly repetitive sequences, we also performed similar analyses on *Minc* IsoSeq transcriptome data from mixed developmental stages retrieved from the NCBI (BioProject PRJNA787737, [https://www.ncbi.nlm.nih.gov/bioproject/PRJNA787737/]). We identified 22 isoseq reads that contained at least one of the three motifs and mapped them to the *Minc* genome (Supplementary Data 9). One long (2045 bp) read aligned on its whole length on contig 24 of the *Minc* genome. The alignment was on the negative strand at the beginning of the contig in a region spanning two repeated units of the *Minc* telomeric sequence and the terminal region of a protein-coding gene (Minc_v4_shac_contig_24g0207501) just upstream of the repeat array. Another shorter read (293 bp), containing motif-3 in full length as well as motifs 2 and 1 in almost full length, mapped at terminal parts of contigs 1, 79, and 47 in the *Minc* genome in regions corresponding to the repeat arrays. Three other long-reads contained two out of three motifs and were also mapped to telomeric regions.

We also searched for transcriptional support of the candidate telomeric repeats in *Mjav* and *Mare* using de novo assembled RNA-seq data from a mixture of eggs and J2s generated on the same populations in a previous study[20]. In *Mjav*, we found 116 isoforms, from 23 transcripts containing at least one telomeric motif. From these, the MjM2 was the motif with the most matches (53), followed by MjM1 (44), and MjM3 (37). Fifteen isoforms contained both MjM1 and MjM3, which are

**Table 3 | Number of transcripts assembled from short-reads (including isoforms) containing telomeric motifs and their length distribution in parentheses**

| Species | Stage / Motif | Motif-1 | Motif-2 | Motif-3 |
|---------|---------------|---------|---------|---------|
| *Minc* | egg | 4 (209–222) | 11 (219–764) | 4 (222–397) |
| *Minc* | J2 | 5 (213–430) | 5 (222–262) | 30 (647–2690) |
| *Minc* | J3-J4 | 1 (227) | 4 (203–230) | 2 (214–252) |
| *Minc* | female | 3 (204–286) | partial (N/A) | partial (N/A) |
| *Mare* | egg + J2 | 54 (209–809) | 64 (203–809) | 59 (203–809) |
| *Mjav* | egg + J2 | 44 (205–928) | 53 (201–628) | 37 (208–461) |

Partial means only partial match with the motifs passing the e-value thresholds.

consecutive on the *Mjav* consensus repeat. For 19 of the 23 transcripts, at least one isoform could be mapped to the *Mjav* genome with 94% of identity and at least 80% of their length, (Supplementary Data 10). In *Mare*, we found 116 isoforms from 14 transcripts that contained at least one motif constitutive of the repeat. The motif with the most matches was MaM2, with 64 isoforms, followed by MaM3 (59 isoforms) and MaM1 (54 isoforms). For all the 14 transcripts, at least one predicted isoform mapped on the genome with at least 94% identity on 80% of its length (Supplementary Data 11).

As for *Minc* short-read data, no assembled isoform of *Mjav* contained all three motifs. However, we could find two isoforms from the same assembled transcript of *Mare* containing the three motifs. Long-read transcriptome data for these two species will be necessary to further characterize these transcripts in the future.

### The repeat has telomeric localization at one single extremity of most chromosomes in the three species

The candidate telomeric repeats we have identified in the three *Meloidogyne* genomes are stranded at contig extremities, relatively G-rich, predicted to form G4 structures and with evidence for transcription, similarly to TERRA in other animals. Although these features are reminiscent of known telomeres in eukaryotes, it must be confirmed that these repeats are actually present at chromosome extremities.

To determine the actual localization of the identified repeats on *Minc, Mjav*, and *Mare* chromosomes, we performed FISH experiments. Specific primers were designed on the composite repeat consensus sequences (Supplementary Figs. 11–13). Amplification of the candidate telomeric repeats of each species showed ladder-like organization visible on agarose electrophoresis gel (Supplementary Fig. 17) that is typical for satellite DNA. To localize the repeat on the *Minc* chromosomes, we performed FISH experiments on a total of 10 slides. Although the precise number of chromosomes is difficult to evaluate due to their small sizes, it can be roughly estimated between 45 and 47. This estimation is consistent with a previous analysis on a different strain of *Minc* which counted 46 chromosomes[43] and with an initial meta-analysis which showed the majority of *Minc* populations had between 41 and 46 chromosomes[44].

Even though metaphases are very rarely found, we successfully localized the composite repeat on the chromosomes. Indeed, >50 evaluated nuclei showed the repeat forms arrays at telomeric and subtelomeric regions of almost all *Minc* chromosomes (Fig. 4). We

repeatedly observed only one or two chromosomes without a detectable signal. Detailed analyses showed an unusual distribution pattern of candidate telomeric repeats exclusively at one extremity of most of the chromosomes (Fig. 4, Supplementary Figs. 18 and 19). The signal intensity varied between chromosomes, indicating differences in repeat array lengths of telomeric/subtelomeric sequences. Although it was more difficult to obtain chromosomal preparation in *Mjav* and *Mare* than in *Minc* and the number of chromosomes could not be counted unambiguously the results are similar to those obtained in *Minc* with confirmation of telomeric localization of the repeats and mostly at one side of chromosomes (Supplementary Fig. 20). However, unlike in *Minc*, we systematically observed at least one chromosome with signals at two ends. This ensemble of observations is consistent with our genome assemblies showing a few contigs with the repeat at both ends in *Mjav* and *Mare* whereas none were identified in *Minc*.

To exclude the possibility that the presence of one telomeric region per chromosome is a result of U-shaped chromosomes with overlapped telomeric sequences, we further investigated the telomeric signals in prophase for the three species (Fig. 5). Comparison of signals in both prophase and metaphase in *Minc* showed a similar pattern with about 40–45 signals (Supplementary Fig. 18) which supports the hypothesis that a composite telomeric repeat is mostly located at one end of *Minc* chromosomes. Similarly, the number of signals in *Mjav* and *Mare* is consistent with the number of chromosomes reported previously (42–48 for *Mjav* and 51–56 for *Mare*)[44].

Overall, our results confirm the composite repeat enriched at the extremities of some contigs has a telomeric localization mostly at one extremity of the chromosomes and most likely constitutes unusual and specific telomeric sequences in these species.

## Discussion

Using long-read sequencing, we assembled the genomes of three allopolyploid root-knot nematodes at contiguity levels exceeding Mb scale and unzipped the three AAB (*Minc*) and four AABB (*Mjav, Mare*) sub-genomes. In the three species, we identified different composite repeats enriched mostly at one extremity of several contigs, with one region (motif-1) highly conserved within and across the three species, and two more variable motifs. The repeats possess several characteristic features of known telomeric DNA such as being G-rich, stranded, transcribed, and predicted to form G-quadruplexes. Using FISH, we confirmed the composite repeats have a telomeric localization in the

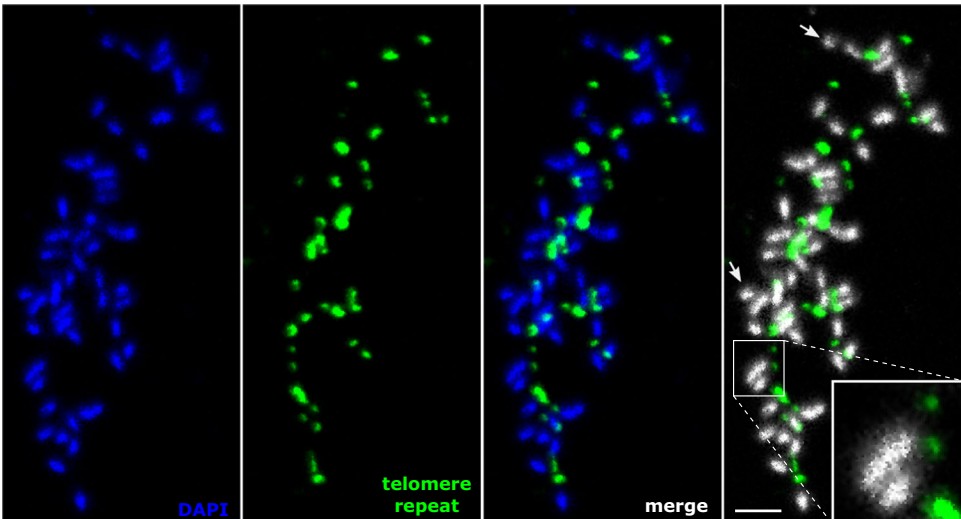

**Fig. 4 | Telomeric chromosomal localization of the *Minc* composite repeat using fluorescence in situ hybridization in metaphase.** Specific primers are used for amplification and labeling of the *Minc* composite repeat enriched at contig extremities in the genome assembly. The FISH experiment was repeated on 10 different microscopic slides. Chromosomes are counterstained with DAPI, arrows represent chromosomes with no visible telomere signal at the end, scale bar = 2 μm.

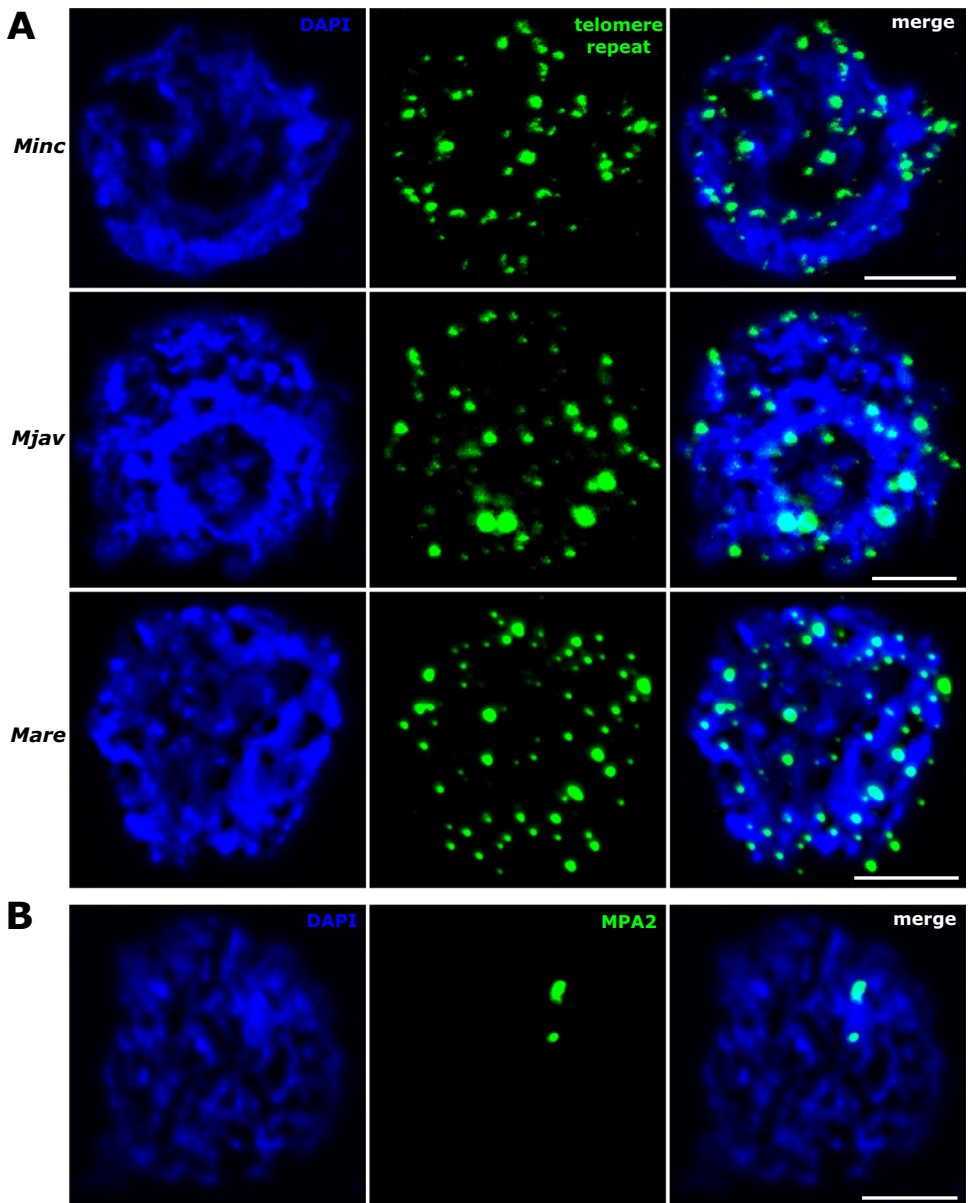

**Fig. 5 | Localization of the telomeric repeats in prophase nuclei. A** Distribution of labeled repeat sequences in *Minc*-*M. incognita*, *Mjav*-*M.javanica* and *Mare*-*M.arenaria*. **B** Labeled probe specific for MPA2[79] sequence and used as a positive control for FISH. For telomere localization, the FISH experiment was repeated on 10 different microscopic slides for each species, and the MPA2 control was stained on 3 slides. Chromosomes are counterstained with DAPI, scale bar = 5 μm.

three species mostly at one extremity of chromosomes and likely represent telomeric DNA. In the context of telomeric repeats evolution, motif-1 was likely subject to selective constraint, while the other parts of the repeating unit (motifs-2 and 3) changed rapidly, creating species-specific profiles of telomeric repeats. Consistent with our findings, similar complex repeats varying from one species to another have been identified at contig extremities on the genomes of the same *Meloidogyne* species in a recent independent study[45].

Although our analysis confirms higher similarity within an A or B sub-genome between species than between sub-genomes within a species, this is not the case for the telomeric repeats. Indeed, each species has a different repeat composition, organization, and density. Moreover, the sequence similarity of conserved repeats within and between species is not consistent with relationships between A and B sub-genomes. We hypothesize that a telomeric variant has spread across all three sub-genomes after hybridization events in each species by concerted evolution. Concerted evolution is characteristic of

tandemly repeated satellite sequences[46]. This is a two-level process in which changes in repeat units are homogenized throughout members of a repetitive family and concomitantly fixed within a group of reproductively linked organisms. As a consequence of concerted evolution, tandemly repeated telomeric families became species-specific, regardless of sub-genomes.

The lack of similarity of the identified repeats with transposable elements suggests they probably constitute satellite DNA. Therefore, the multiplication of these repeats by (retro-)transposition is unlikely. In the absence of telomerase, a likely hypothesis is that they are maintained and multiplied by an alternative lengthening of telomeres (ALT) pathway. ALT involves recombination and has been considered the ancestral telomere multiplication mechanism in eukaryotes[40]. Consistent with an ALT recombination-based mechanism, the telomeric repeats in *Meloidogyne* show heterogeneity in their sequences. Furthermore, we identified TERRA-like transcripts in the three species with the potential to form R-loops

with telomeric DNA while the repeat itself is predicted to form G-quadruplexes. The combined formation of G4s and R-loops is known to result in G-loops which are hypothesized to facilitate ALT recombination[47]. However, in nematodes, it is so far unknown whether RNA from transcribed telomeric DNA plays similarly important roles than TERRA in other animals[42]. Additionally, G-quadruplexes or G-loops could play a role in the recruitment of proteins involved in the maintenance of the telomere integrity. Since no ortholog of any of the single-strand or double-strand telomeric DNA binding proteins of *C. elegans* were found in *Meloidogyne* species, such telomeric proteins remain to be discovered. Indeed, telomere-binding proteins such as those composing the shelterin complex play essential roles in telomere maintenance, stability, and function. In that perspective, it is interesting to note that *Drosophila* species, which also lack telomerase, have replaced the shelterin with another complex called terminin, which plays the same role and is composed of four proteins encoded by orphan genes[41]. It is tempting to speculate that in *Meloidogyne* too, orphan proteins might have been recruited to bind, stabilize, and protect the composite telomeric repeats.

Besides questioning their possible evolutionary origins and the mechanisms involved in their functions, the discovery of these unusual telomeric repeats reveals how chromosomes start (or end) in the most economically important *Meloidogyne* species. This discovery will be pivotal to guide further scaffolding efforts toward full chromosome resolution of these complex genomes. In that perspective, it is interesting to note the consistency between bioinformatics results at the contig level and cytogenetics DNA-FISH observations. Initial cytogenetics analyses performed on hundreds of populations for these and other *Meloidogyne* species suggested the numbers of chromosomes were mostly (41–46) for *Minc*, (42–48) for *Mjav* and (51–56) for *Mare*[44]. In *Minc*, we counted 46–47 chromosomes and found 52 repeat arrays at contig extremities while DNA-FISH revealed 40–45 fluorescence arrays at one end of most chromosomes. This ensemble of observations is consistent with most chromosomes having this repeat at one end. In *Mare*, we identified 57 repeat arrays on 53 contigs, this number is also close to the number of chromosomes (51–56) and compatible with most of them having the repeat at one single end. The situation in *Mjav* is slightly different since 59 repeat arrays were identified in 53 contigs. This is higher than the range of chromosome numbers (42–48) usually observed and compatible with most chromosomes having the repeat at one extremity but a few chromosomes having the repeat at both ends. This ensemble of observations suggests that the repeats we identified are surprisingly at only one end of most chromosomes. Although we identified multiple other contig ends which are repeat-rich and gene-poor, we did not find another evident repeat unit forming arrays in a number close to the number of chromosomes in any species. Therefore, we hypothesize that multiple different repeats, each being specific to a few chromosomes, are present at the other ends and form protective heterochromatin. An alternative hypothesis would be that *Meloidogyne* chromosomes form loops or circular structures. This has been evoked for the uni-chromosomal nematode *D. pachys* which also lacks telomerase[11]. However, no evidence from the genome reads or chromosomal observations supports this possibility at this stage. Further investigations will be necessary to characterize what kind of DNA or structure protects the other ends of most chromosomes.

It is tempting to link karyotypic variations observed in terms of the number of chromosomes in these species to their peculiar telomere systems. Although differences in repeat numbers and densities are observed between the three *Meloidogyne* species, similar levels of karyotypic variations are observed. Furthermore, we note that diploid meiotic root-knot nematodes (i.e., *M. hapla*, *M. graminicola*, and *M. chitwoodi*) also lack telomerase and canonical telomeric repeat but have more stable karyotypes. Consequently, the unusual telomere system of root-knot nematodes might still be efficient in preventing chromosomal fusions. Instead, polyploidy and lack of selective pressure for the pairing of homologous chromosomes combined with their holocentric nature[43] might be the main reason for karyotypic plasticity in these allopolyploid species.

Overall, the composite telomere repeats we identified in these three devastating and allopolyploid *Meloidogyne* species show conservation yet variations between these species and completely lack recognizable homology in any other species represented in public sequence libraries. Therefore, targeting the telomeres of these *Meloidogyne* species theoretically presents the double advantage of potentially highly detrimental effects on the survival of the parasites with unlikely unintentional effects on the other species, opening interesting perspectives toward the development of innovative control strategies.

## Methods

### *Meloidogyne* samples, DNA extraction, library preparation and sequencing

The same *Minc*, *Mjav* and *Mare* isolates previously sequenced with short-reads[20] and maintained in the INRAE collection were reared on tomato plants. For each isolate, eggs were collected seven weeks after infection and DNA was extracted using the Epicentre® MasterPure™ Complete DNA and RNA Purification Kit (Lucigen, Biosearch Technologies) with minor modifications to be gentler and protect the DNA molecule: centrifugation steps were reduced to 3000 × *g* for a longer period of 20 min (instead of ≥10,000 × *g* for 10 min) and pipetting was kept to the strict minimum.

A total of 11, 8, and 9 ONT libraries for *Minc*, *Mjav*, and *Mare*, respectively, were prepared using the SQK-LSK108 and SQK-LSK109 ligation sequencing kits, following kit changes and Oxford Nanopore instructions over time (Supplementary Data 12). Libraries were loaded on MinION or PromethION R9.4 Flow Cells, according to the Oxford Nanopore protocols, targeting over 100X coverage per genome.

To allow contigs polishing, PCR-free Illumina libraries were prepared using the Kapa Hyper Prep Kit (KapaBiosystems, Wilmington, MA, USA), following the manufacturer's instructions. Libraries were then quantified by qPCR using the KAPA Library Quantification Kit for Illumina Libraries (KapaBiosystems), and library profiles were assessed using a High Sensitivity DNA kit on an Agilent Bioanalyzer (Agilent Technologies, Santa Clara, CA, USA). Libraries were then sequenced on an Illumina NovaSeq instrument (Illumina, San Diego, CA, USA), using 250 base-length read chemistry in a paired-end mode, generating approximately 100X coverage for each genome (Supplementary Table 3).

### Genome assembly

**ONT reads QC and filtering.** We first used guppy-v6.0.6 in super high-accuracy mode with a minimal quality score of 7 (--min_qscore 7) to base-call reads from the raw FAST5 sequencing signals. Raw sequencing libraries and the configuration files used for the base-calling are described in Supplementary Data 12. We then analyzed the reads quality and length distribution as a quality check using cONTent, a homemade script available at https://github.com/DjampaKozlowski/cONTent.

Following QC, we used nanofilt[48] to crop the 50 first and the 30 last nucleotides as these regions showed reduced per-base quality. We then selected reads that were at least 2 kb long and had a quality score Q > 12. For *Minc*, this returned 1,709,063 higher-quality cleaned reads out of a starting number of 6,619,891 raw reads. The same method was used for *Mare* and *Mjav*, yielding 2,592,069 and 3,144,046 higher-quality reads from 10,506,836 and 11,506,089 raw reads, respectively (Table 4).

**Illumina reads QC and filtering.** Low-quality nucleotides (Q < 20) from both ends of the reads that passed Illumina filtering were discarded. Illumina sequencing adapters and primer sequences were removed from the reads. Then, reads shorter than 30 nucleotides after

trimming were discarded. These trimming and removal steps were achieved using in-house-designed software based on the FastX package (https://github.com/institut-de-genomique/fastxtend). The last step identifies and discards read pairs that are mapped to the phage phiX genome, using SOAP aligner[49] and the Enterobacteria phage PhiX174 reference sequence (GenBank: NC_001422.1). This processing, described in[50], resulted in high-quality data. After all these steps, reads shorter than 167 bp were eliminated.

**Assembly.** We assembled the genomes from cleaned ONT data using NECAT[51] (version 20200119) with default parameters and an input genome size of 200 Mb for *Minc*, 310 Mb for *Mare*, and 300 Mb for *Mjav*. These input genome sizes do not influence final assembly size but are used to sample the longest highest quality long-read covering the genome at at least 40X. After *Minc* assembly and bridging, NECAT formed 298 contigs for a genome size of 199,975,050 bp and an N50 length of 1,856,931 (longest contig = 4,672,579 bp, L50# = 37). For *Mare*, we obtained 398 contigs with an assembly size of 305,105,593 bp and a N50 of 2,046,842. For *Mjav*, we obtained 371 contigs with an assembly size of 298,215,187 bp and a N50 of 2,079,353.

**Polishing and contamination check.** We performed three successive polishing steps for each species. Using the assembly produced by NECAT, we ran two rounds of Racon[52] (version 1.4.10) using the cleaned ONT reads. The resulting fasta file from the first round was used as a reference genome for the second round of mapping and correction. For each species, between 6 (*Mjav*) and 20 (*Mare*) contigs which did not have enough coverage by the long-reads, were dropped. The new set of polished contigs from Racon were further polished by one round of Medaka (version 1.4.4) (https://github.com/nanoporetech/medaka). Finally, the contigs polished with long-reads were further polished with cleaned PCR-free Illumina short-reads using one round of Hapo-G[53] (version 1.2) with default parameters. This led to polished assemblies with 293, 378 and 365 contigs for *Minc*, *Mare* and *Mjav*, respectively (Table 4). We then use blobtools to detect possible contaminations based on contig GC content, coverage and taxonomic assignment[54]. The analysis revealed no contamination but allowed identifying contigs that corresponded to the mitochondrial genomes (Supplementary Fig. 4). These contigs were removed from the final assembly, and the final number of contigs was 291, 377 and 364 for *Minc*, *Mare* and *Mjav*, respectively (Table 4).

**Genome ploidy, divergence, and completeness assessment**
We used JellyFish[55] (version 2.3.0) with the default k-mer size of 21 to enumerate k-mers on the cleaned short-reads for each species. Using the JellyFish counts, we generated histograms using the jellyfish histo command and adjusted the -h parameter to allow k-mer repeated up to 1 million times to make sure downstream analysis will not be limited by the default maximum allowed occurrence of k-mers. We then used SmudgePlot[56] to infer the ploidy level for each species (Supplementary Fig. 1). Finally, we used GenomeScope2[56] with the previously identified ploidy levels and generated k-mer histograms to estimate genome sizes and average percent of nucleotide divergence between homoeologous genome copies (Supplementary Fig. 2).

We estimated genome completeness using two strategies. First, we used KAT[57] to estimate whether the diversity of k-mers present in the cleaned short-reads was represented in the genome assemblies for each species (Supplementary Fig. 3). Second, we used CEGMA[23] and BUSCO[24] to estimate which percentage of evolutionarily conserved single-copy genes could be retrieved in the genome assemblies. We used CEGMA version 2.5 with default parameters to identify eukaryotic evolutionarily conserved genes and their average numbers in the *Meloidogyne* genomes. We used BUSCO version 5.4.4 with the Metazoa odb9 dataset and default parameters to estimate the percentage of widely conserved animal genes retrieved in complete and partial length in the three genomes.

**Gene predictions, duplications detection and classification**
Predictions of gene models in *Minc*, *Mjav* and *Mare* genomes were done with the fully automated pipeline EuGene-EP version 1.6.5[25]. EuGene has been configured to integrate similarities with known proteins of *C. elegans* (NCBI BioProject PRJNA13758, available at [https://www.ncbi.nlm.nih.gov/bioproject/PRJNA13758]) downloaded from Wormbase ParaSite[58] as well as the *Nematoda* section of Uni-ProtKB/Swiss-Prot library[59], with the prior exclusion of proteins that were similar to those present in RepBase[39]. We used as transcriptional evidence, transcriptome data for *Minc*, as it is the *Meloidogyne* species with the most comprehensive expression data available. RNA-seq data from pre-parasitic J2, J2-J3 and adult female stages[20] were assembled de novo using Trinity[60] followed by a cleanup that retains for each trinity locus only the transcript that gives the longest ORF. The dataset of *M. incognita* assembled transcriptome was aligned on the genomes of the four *Meloidogyne* species using Gmap[61] and except for *Minc* the 'cross-species' option was used. Only alignments spanning 30% of the transcript length with at least 97% identity were retained. Statistics of gene predictions in the four species are available in Supplementary Table 1. The EuGene default configuration was edited to set the 'preserve' parameter to 1 for all datasets, the 'gmap_intron_filter' parameter to 1, the minimum intron length to 35 bp, and to allow the non-canonical donor splice site GC. Finally, the nematode-specific Weight Array Method matrices were used to score the splice sites (available at this URL: http://eugene.toulouse.inra.fr/Downloads/WAM_nematodes_20171017.tar.gz).

To identify the conserved syntenic regions within and between each genome (*Minc*, *Mjav*, and *Mare*), we used McScanX[26] with default parameters. For each species, we used as input to McScanX an all-against-all BLASTp comparison of the proteins and the position of the protein-coding genes in the genome. Using the resulting collinearity files from McScanX, we determined the rates of non-synonymous (Ka) and synonymous (Ks) mutations of each gene-pair with the script add_kaks_to_MCScanX.pl and formatted the result using the script calculate_collinarity_metric.pl, both available at https://github.com/reubwn/collinearity/tree/master.

**Assigning A/B sub-genomes to the contigs**
We used the conserved synteny and Ks value results to assign A or B sub-genomes to the contigs of the three *Meloidogyne* species. For *Minc*, based on the collinearity results from McScanX, we selected only triplicated synteny blocks on different contigs. All the triplicated blocks belonging to the same triplet of contigs were concatenated per contig to create one single bigger block. For each concatenated block (of at least 10 genes in a row), we retrieved the Ks values for each gene-pair comparison, and we calculated the median value for each pairwise

**Table 4 | Raw statistics of genome assembly based on ONT long reads**

| Species | raw reads | clean reads | reads used by Necat | contigs after polishing | contigs w/o mitochond. | genome size (Mb) |
|---------|-----------|-------------|---------------------|-------------------------|------------------------|------------------|
| *Minc* | 6,619,891 | 1,709,063 | 287,593 | 293 | 291 | 199.42 |
| *Mare* | 10,506,836 | 2,592,069 | 447,241 | 378 | 377 | 304.33 |
| *Mjav* | 11,506,089 | 3,144,046 | 279,109 | 365 | 364 | 297.81 |

contig comparison. All the values assigned as −2 or superior to 1 were ignored from the analysis as they respectively represent cases where a Ks value could not be calculated or saturated rate of synonymous mutations. Because *Minc* has an AA'B structure, for all pairwise comparisons of median Ks between triplets of contigs, we considered the lower median Ks value as representative of the divergence between A and A' and the two higher median Ks values as representative of the divergence between B and the two As. Using this rule, it was possible to separate B from As in the triplets. In case a contig belongs to several triplets, due to the fragmentation of the genome, we used a majority rule over all the pairwise comparisons to assign a B or A sub-genome to that contig. In case of equal numbers of A and B assignments we preferred considering the contig not assigned and it was tagged as N/D (Not Determined). The contigs with no collinear blocks were also assigned as N/D (Not determined). Although this strategy allowed to separate B from the As sub-genomes, it was not possible to further determine which contig was A and which was A' because they were equidistant from B.

For the two tetraploid species, we extended the same strategy to quadruplets of contigs rather than triplets. Because the divergence between the two copies of AA' and BB' are similar, we can only separate the contigs into two different groups according to the higher Ks values. Then to assign an A or B sub-genome, to *Mjav* and *Mare* contigs, we used comparative analysis with *Minc* contigs (see below).

To achieve this, we performed a new cross-species synteny analysis with McScanX, using both the comparison between *Minc* and *Mare*, and *Minc* and *Mjav*. We selected all groups of conserved synteny between the species that were in three copies in *Minc* and four copies either in *Mjav* or *Mare*. After calculating the median Ks values for each group of seven contigs (three contigs from *Minc* and four for *Mjav/Mare*), we performed a clustering analysis, using Euclidean distance in R, to produce classification trees of the contigs according to within and between-species median Ks values. Using this strategy, we could assign A and B subgenomes to *Mjav* and *Mare* contigs according to their closest relationship in the classification tree to either an A or B contig of *Minc*. We plotted the tree topologies, and we visualized them on iToL[62]. All the scripts used for the assignment of subgenomes in the three *Meloidogyne* species are available at https://github.com/azotta/MeloidogyneGenomes.

### Transposons and other repetitive elements annotation

We predicted transposable and other repetitive elements on the three genome assemblies using EDTA[38] version 2.1. To allow a more comprehensive annotation of repetitive elements, we used the –sensitive parameters, which use RepeatModeler[63] annotations to identify other repeats that were not identified previously. We configured RepeatModeler to identify the remaining repeats (sensitive 1), to perform a whole genome repeat annotation (anno 1), and to evaluate the consistency of the previous annotation (evaluate 1), the rest of the parameters were left to default.

### Distribution of telomeric repeats and telomere-associated proteins in nematode genomes

**Dataset of nematode genomes and proteomes.** We collected predicted nematode proteomes, and genome assemblies with a minimal N50 length of 80 kb. We started by the list of sequenced genomes at WormBase ParaSite[58] and within a given genus we kept up to three genomes with the highest N50 lengths. Then, we complemented this list with the ensemble of nematode genomes and proteomes available at the NCBI, using the same criteria for selection. Our selection included the *C. elegans* genome and proteome and we did not apply a limitation of 3 species for the *Meloidogyne* genus, since this was the focal point of our study. Because no gene predictions were available for *Meloidogyne luci*[34] despite its genome satisfying the N50 threshold, we predicted gene models using the same strategy as used for *Minc*, *Mjav* and *Mare*

with Eugene[25] and with the *Minc* transcriptome as a source of evidence. To anchor our analysis, we included the genome assembly and predicted proteins of the tardigrade *Hypsibius exemplaris*[64]. This resulted in 69 nematode species and one outgroup (Supplementary Data 1).

**Detection of homologs of *C. elegans* telomere-associated proteins.** Starting from *C. elegans* proteins known to be associated to telomeres, including the telomerase Trt-1 as well as telomere-binding proteins Pot-1, Pot-2, Pot-3, Mrt-1, Tebp-1/Dtn-1 and Tebp-2/Dtn-2, we searched for homologs in the other nematode proteomes and genomes selected above. In addition to these proteins, we also included Clk-2 and Mrt-2 which have suspected roles in telomere maintenance but are also important DNA damage checkpoint proteins. We employed the following strategy:

(i) we retrieved protein motifs characteristic of the *C. elegans* proteins under consideration and used hmmsearch from HMMER3[65,66] to determine their presence in the predicted proteomes of all the collected species. We set an e-value threshold allowing retrieval in the *C. elegans* proteome of the query protein and no unrelated proteins. For TRT-1, we used the 'telomerase reverse-transcriptase' PANTHER[32] domain (PTHR12066) with an e-value of $1E^{-15}$. POT-1 presented no specific protein-domain, and this domain-based strategy could not be used for this protein. The single-strand telomeric DNA-binding proteins POT-2, POT-3 as well as MRT-1 all possess a 'ssDNA-binding domain of telomere protection protein' Pfam[67] domain (PF16686) which was used as a query with an e-value of $1E^{-15}$. For MRT-2, we used the 'Repair protein RAD1/REC1/RAD17' Pfam domain (PF02144) as a query with an e-value threshold of $1E^{-40}$. For the double-strand telomeric DNA-binding proteins TEBP-1 and TEBP-2, we used as query the Panther domain 'GA binding and activating and spk (spk) domain containing-related' (PTHR38627) with an e-value threshold of $1E^{-10}$. Finally, for CLK-2, we used the Panther domain PTHR15830 'Telomere length regulation protein Tel2 family member' with an e-value threshold of $1E^{-15}$.

(ii) we performed an Orthofinder[68] analysis of all the proteomes (69 nematodes +1 tardigrade) to classify the different proteins in orthogroups. We used the option -M msa to calculate multiple sequence alignments and phylogenies for all the orthogroups. Then, for each *C. elegans* protein of interest we retrieved the corresponding orthogroup and checked which other species were present in the same orthogroup and thus had a putative ortholog. For the specific case of telomerase we also used the *Ascaris suum* enzyme as a query, because strong evidence for an active telomerase in this species has been reported[69].

(iii) in case of absence of evidence for a homolog or disagreement between (i) and (ii) we retrieved the *C. elegans* protein sequence for the gene of interest and aligned it in against nematode genomes translated in the 6 frames using tBLASTn with an e-value threshold of 1E-5. We also repeated the same tBLASTn search against WormBase Parasite[58] to extend the search to different versions of a genome assembly for a given species.

For each species, homologs of *C. elegans* telomere-associated genes were considered present if evidence from (i) and (ii) could be identified. In case of disagreement between (i) and (ii), but significant tBLASTn matches in (iii), homologs were considered possibly present but misannotated. If no evidence from either (i), (ii) or (iii) could be identified, the gene was considered as having no ortholog in the considered species. If only evidence for either (i) or (ii) were present without confirmation by (iii), we considered the presence of an ortholog unsure.

**Detection of canonical nematode telomere repeats.** The 69 nematode genomes were scanned for the canonical (TTAGGC)n nematode

repeat using the Telomere Identification toolKit Tidk (https://github.com/tolkit/telomeric-identifier). We used the command Tidk find with -c nematoda option and a window size of 1000 nucleotides. The nematode telomere repeat was considered present if repeated at least 50 times within the 2000 last or first nucleotides of at least 3 contigs. Because repetitive regions might be misassembled in the genomes we also used fuzznuc from the EMBOSS package[70] to further search $(TTAGGC)_2$ dimers anywhere on the genome assemblies. We considered the telomeric repeat possibly present if repeated at least 100 times (50 consecutive dimers). Finally, the absence of the canonical $(TTAGGC)_n$ repeat may also be due to repeat filtering during the genome assembly process. Therefore, the canonical telomeric repeat was searched in raw Illumina reads whenever available in SRA. For each species, when available, we chose the SRA data produced with illumina NovaSeq for WGS. We used $(TTAGGC)_{10}$ as a query to perform a blastn search on NCBI, using the SRA library corresponding to the species of interest as a subject and deselecting low complexity filtering. We considered the telomeric repeat as possibly present when more than 100 matches of $(TTAGGC)_{10}$ could be retrieved in the reads. The canonical repeat was considered present in a species whenever evidence from either Tidk or from fuzznuc plus blast against SRA could be retrieved. When evidence from fuzznuc was found but no SRA data was available the canonical repeat was considered possibly present. In all other cases, it was considered absent.

**Projection of the results on a tree and reconstruction of ancestral states.** We used a recently published phylogenomic analysis of the nematode phylum[35] as a reference to draw a phylogenetic tree of the 69 nematode species and added the tardigrade *H. exemplaris* as an outgroup. Based on the above-mentioned rules, we generated a three-state matrix of presence (1) absence (0) and unsure presence (0/1) of the TTAGGC repeat as well as the telomerase and other telomere-associated proteins in each species. We used iTOL[62] to display these data on the tree and Mesquite[71] to reconstruct ancestral states using maximum parsimony.

### De novo identification of candidate telomeric repeats
We searched for clusters of any perfect repeats of sizes 6–12 using the equicktandem and etandem commands from the EMBOSS package[70]. Because no simple tandem repeats of size 6–12 were identified at contig extremities, we searched for more complex and possibly degenerated motifs.

For each species, we iteratively extracted the first and last 2 and 5 kb of all contigs and searched for frequent motifs using MEME[72] in classic mode on these sequences. Any number of repetitions (anr) option was selected with iteratively searching motifs from size 50 to 100 nucleotides, allowing up to 5 different motifs reported. After examination of MAST results from the MEME suite, we identified the most frequently co-occurring motifs in the contig extremities (either 2 or 5 kb). We then only selected those contig beginnings and ends that contained repeats of these co-occurring motifs and launched a second round of MEMEmotif search. This new motif search was with a maximum of 3 motifs with a size range of 80–160 bp. The results of MAST allowed the identification, for each species (*Minc*, *Mjav*, and *Mare*), of a suite of 3 enriched motifs.

For each species, we considered the suite of 3 motifs as a repeat unit and extracted all these repeat units in the corresponding genomic regions. We aligned these repeat units using MAFFT v7[73] with option G-INS-i. The obtained multiple sequence alignment was then used as input to the EMBOSS[70] Cons program to build a consensus.

We also retrieved for each enriched motif of each species, the corresponding FASTA sequences from MEME results, built a multiple alignment with MAFFT using the same parameters as for the whole repeat unit, and constructed a hmm profile for each motif, using hmmbuild from the HMMER package[66].

Finally, we compared the 9 motifs (3 species ×3 motifs) all-against-all, using two different approaches. (i) Using TomTom from the MEME suite, we aligned all the MEME motifs against each other and retrieved the e-values of these alignments. We used these e-values to produce a distance matrix with the Ward method and produced a hierarchical clustering dendrogram in R. (ii) We used MAFFT with the –auto option to build a multiple alignment of the consensus sequences of the 9 motifs. We used this alignment as an input for IQ-tree[74] to produce a maximum likelihood phylogenetic tree with automatic detection of the most appropriate evolutionary model and 1000 fast bootstrap replicates. The two methods returned the same topology and classification in three clades.

To verify the existence of other complex motifs on the other end of contigs, we first calculated the average length of the telomeric regions found for each species (16 kb for *Minc*, 10 kb for *Mjav*, and 20 kb for *Mare*). We iteratively extracted both the beginnings and ends of all contigs with the corresponding average length and calculated the number of coding bases (belonging to CDS) and repetitive bases (belonging to an EDTA repeat). We calculated the ratio of repetitive bases / coding bases. Based on the values obtained for the regions with telomeric sequences already identified, we could identify other contigs with similar values. We retrieved these regions and performed the MEME analysis as described above.

### Distribution and comparison of the newly identified candidate telomeric repeats in *Meloidogyne* and other genomes
For each species, we used the consensus sequence of the composite repeat as a query in a blastn[75] search against their respective genome with an e-value threshold of 1e−35 and no dust filter. We sorted the blastn results per position on the contigs to produce a table of their distribution. We also used individual HMM profiles of each of the 3 constitutive motifs separately and searched them against the genomes using nhmmer[66] with e-values of 1e−23 for the highly conserved motif-1 and 1e−18 for the more degenerated motifs 2 and 3.

To compare motif-1 within and between species we retrieved all the occurrences identified by the nhmmer search against the respective genomes. We then used CD-HIT[76] to cluster them with an identity threshold of 99.9% and a coverage threshold of 90%.

To investigate whether the three consensus sequences of the repeat units were conserved in other nematodes, we also searched with blastn and with the same parameters in all the other nematode genomes we downloaded in this study (Supplementary Data 1). Furthermore, for each of the three composite repeat units, we also performed an online blastn search against all the nematode genomes present in Wormbase ParaSite[58] with an e-value of 0.01 and no low complexity filter.

Finally, we also investigated more widely whether the candidate complex telomeric repeats had homologs against the NCBI's nt library using blastn online with an e-value of 0.01 and the low complexity filter deactivated.

Besides the whole repeat unit, we also investigated how each individual motif constituting them was conserved across *Meloidogyne* and other species. The hmm profile of each motif was searched against the 69 nematode genomes we collected using nhmmer[66]. For the non-degenerated motifs-1 of the three species, we used an e-value of 1e−23 while for the more degenerated motifs 2 and 3, we used an e-value of 1e−18.

### G-quadruplex prediction
We used the G4-Hunter software[77] with a window size of 25 and a minimal score of 1.2 to detect regions possibly forming G-quadruplexes on the composite terminal repeats as well as the whole genome sequences of the three *Meloidogyne* species. Using an in-house script, we retrieved only the non-overlapping positions, distant by more than 25 bp, and the score for each G-quadruplex position, and we plotted their frequency using Rstudio.

## Transcriptomic support of telomeric repeats

For *Minc*, we retrieved RNA-seq Illumina data of four developmental life stages (eggs, pre-parasitic J2, J3-J4 and adult female) from a previous publication[20] (BioProject PRJEB8846, [https://www.ncbi.nlm.nih.gov/bioproject/PRJEB8846]). For *Mjav* and *Mare*, we retrieved from the same publication the RNA-seq data from a mix of eggs and J2 juveniles J2 (BioProject PRJEB8843, [https://www.ncbi.nlm.nih.gov/bioproject/PRJEB8843] and BioProject PRJEB8845, [https://www.ncbi.nlm.nih.gov/bioproject/PRJEB8845]). For each *Minc* developmental life stage, we pooled the triplicates to have one dataset per life stage. For the three species, we performed a de novo transcriptome assembly using Trinity-v2.4.0[60]. Then, for each species, we used nhmmer[66] with the hmm profiles of each motif as queries to identify assembled transcripts and isoforms containing one or several of these motifs. We repeated the same nhmmer search on an Iso-seq mixed-stage *Minc* transcriptome downloaded from the NCBI (BioProject PRJNA787737, [https://www.ncbi.nlm.nih.gov/bioproject/PRJNA787737/]). Assembled transcripts as well as Iso-seq reads that contained one or several telomeric motifs were aligned on the corresponding genomes using GMAP[61] with default parameters, and -S option for summary output.

## Confirmation of telomeric position of the identified repeats with FISH

Specific primers (Supplementary Table 4, Supplementary Figs. 11–13) were designed on the consensus candidate telomere sequence for each species (*Minc, Mjav, Mare*) using Primer 3 software[78]. Probes for FISH experiments were prepared by PCR labeling with biotin-16-dUTP (Jena BioScience) using genomic DNA as template. The PCR was prepared in 25 μL volume containing 1x Colorless GoTaq Flexi Buffer, 2.5 mM MgCl2, 0.1 mM dNTPs, 0.4 μM primers, 1.25 U of GoTaq DNA Polymerase (Promega) and 0.02 ng of gDNA. For PCR settings 3 min at 95 °C of initial denaturation was followed by 35 amplification cycles (20 s at 95 °C, 20 s at 62 °C-*Minc*, 51.5 °C-*Mjav*, 55.4 °C-*Mare* and 40 s at 72 °C) and 5 min at 72 °C for final extension. The prepared probes were cleaned using QIAquick PCR Purification Kit (Qiagen) and eluted in 30 μL of nuclease-free water and checked on 1% agarose gel for successful biotin incorporation (Supplementary Fig. 17). The corresponding labeled probes had lengths (200–400 bp) ideal for hybridization steps during in situ localization.

Microscopic slides and subsequent fluorescent in situ hybridization (FISH) were performed as described in ref. 43. Briefly, cell suspension from isolated females infecting tomato roots was applied on slides using Cytospin 4 cytocentrifuge (Shandon, ThermoFisher Scientific). Slides were incubated for 20 min at −20 °C in methanol:acetone (1:1) fixative, dried, and carried through the FISH. Specimens are pretreated in 45% acetic acid, incubated for 30 min at 37 °C with RNase A, and fixed with 1% formaldehyde in PBS with 50 mM MgCl2. Chromosome denaturation was performed in 70% formamide in 2xSSC at 70 °C for 2 min. On each slide, 100 ng of lyophilized labeled probe resuspended in hybridization buffer and denatured at 75 °C for 5 min was applied and incubated at 37 °C overnight. Afterwards, 4× post hybridization washes in 50% formamide in 2xSSC and immunodetection with fluorescein avidin D and biotinylated antiavidin D system (Vector Laboratories) through three layers of signal amplification with fluorophore conjugates and antibodies. For dilutions, 1:500 fluorescein avidin D, 1:100 biotinylated anti-avidin D, and 1:2000 fluorescein avidin D were used. Slides were counterstained with DAPI, mounted with Mowiol 4-88 (Sigma-Aldrich), and images recorded using a confocal laser scanning microscope Leica TCS SP8 X. For each cytosmear 6–8 z-stacks slices with an average thickness of 5 μm were acquired and images were post-processed using ImageJ and Adobe Photoshop software. Telomeres were quantified on the separated green channel, which was automatically thresholded, and spots were counted based on their mutual separation using the Analyze Particles tool in ImageJ.

## Reporting summary

Further information on research design is available in the Nature Portfolio Reporting Summary linked to this article.

## Data availability

All the raw long and short genome sequencing reads for the three species have been deposited in the EBI's European Nucleotide Archive (ENA) under BioProject PRJEB61149 with detail of each library and accession numbers in Supplementary Data 12 and Table 3. RNA-seq raw data used for transcriptome assembly of Minc, Mjav, and Mare are under BioProject PRJEB8846 (Minc RNA-Seq raw data), BioProject PRJEB8843 (Mjav RNA-Seq raw data), and BioProject PRJEB8845 (Mare RNA-Seq raw data). Genome and transcriptome assemblies as well as annotations have been deposited in the French national data repository 'Recherche Data Gouv' and are publicly accessible at this address: https://entrepot.recherche.data.gouv.fr/dataverse/Melo-Telo/. The ISO-seq data was retrieved from BioProject PRJNA787737. The source data for Figs. 1 and 3 and Supplementary Figures 14, 15, and 16 are provided as a Source Data file. Source data are provided with this paper.

## Code availability

All the custom codes and scripts used in this study are publicly available on GitHub without restriction for re-use with links to the resources cited at the appropriate position in the manuscript. All already published codes are cited by their references. cONTent [https://github.com/DjampaKozlowski/cONTent] (ONT reads quality and length distribution quality check). Medaka [https://github.com/nanoporetech/medaka] (ONT polish software). Collinearity [https://github.com/reubwn/collinearity/tree/master] (ka/ks result from MCscanX formatting). AssignGenomes [https://github.com/azotta/MeloidogyneGenomes] (Assign the A and B genomes of *Meloidogyne*). TIDK [https://github.com/tolkit/telomeric-identifier] (Canonical Telomeric sequence identification).

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

## Acknowledgements

We are grateful to the genotoul bioinformatics platform Toulouse Occitanie (Bioinfo Genotoul, https://doi.org/10.15454/1.5572369328961167E12) as well as OPAL infrastructure and the Université Côte d'Azur's Center for High-Performance Computing for providing computational resources and support. We are grateful to the bioinformatics and genomics platform, BIG, Sophia Antipolis (ISC PlantBIOs, https://doi.org/10.15454/qyey-ar89) for computing and storage resources. We would like to thank Eric Gilson, Marine Poullet, Carole Belliardo, Julia Truch, Philippe Castagnone-Sereno, Pierre Abad, and Claire Caravel for help, support and stimulating discussions. We are grateful to Paulo Vieira and Sebastian Eves-van den Akker for providing early access to the genome and predicted proteins of *Pratylenchus penetrans*. This work was supported by the Genoscope, the Commissariat à l'Énergie Atomique et aux Énergies Alternatives (CEA) and France Génomique with a 'Grand Projet de Séquençage' grant ALPAGA attributed to E.G.J.D (ANR-10-INBS-09-08). A.-P.Z.M. and G.D.K were financially supported by the French government, through the UCAJEDI Investments in the Future project managed by the National Research Agency (ANR) under reference number ANR-15-IDEX-01. A.-P.Z.M., L.P-B, E.G.J.D, E.D-S. and N.M. received support from the France-Croatia bilateral cooperation initiative COGITO (49274ZB).

## Author contributions

A.P.Z.M. assembled and polished the genomes, checked for contamination, identified the telomeric repeats in *Mjav* and *Mare*, performed repeat and transposons analysis, searched canonical telomeric repeats in Illumina short-reads repositories, searched G-quadruplex structures in telomeric motifs and genomes, analyzed iso-seq transcriptome data, identified and separated A and B sub-genomes, participated in writing the manuscript. G.D.K. participated in genome assembly and analysis, identified the telomeric repeat in *Minc*, searched canonical telomeric repeats in Illumina short-reads repositories, aligned and analyzed *Minc* transcriptome data from de novo assembled short-reads. L.P-B. designed and optimized the protocols, performed, and supervised the DNA extractions for genome sequencing, wrote the corresponding material and methods section. E.D-S. performed fluorescence in situ hybridization experiments, analyzed and interpreted the results, and participated in writing the manuscript. K.L. designed the protocols, performed, and supervised ONT and Illumina sequencing of genomic DNA. J-M.A. participated in designing the genome assembly strategy, managed genome data deposition, suggested edits in the manuscript. K.R-S. searched canonical nematode telomeric repeats in nematode genomes, listed telomere-associated proteins in *C. elegans* and their functions, and participated in writing the manuscript. M.B-B. performed analysis of co-distribution of telomeric repeats and regions forming G-quadruplexes in the *Meloidogyne* genomes, designed the graphical representation, analyzed the results, and proposed edits in the manuscript. C.B. participated in ONT and Illumina sequencing of genomic DNA, wrote the corresponding methods sections. A.P. searched canonical nematode telomeric repeats in nematode genomes, performed BUSCO evaluations of genome assemblies. C.R. performed gene predictions in the *Minc*, *Mjav*, *Mare* and *M. luci* genomes, wrote the corresponding methods section. D.K.K. re-basecalled all the ONT reads in super-high accuracy mode, participated in designing the repeat analysis strategy, performed ploidy, heterozygosity and genome sizes estimations based on k-mers. R.H-G. performed DNA extractions for genome sequencing. M.D.R. assembled de novo the *Minc* transcriptomes based on RNA-seq data and analyzed them, participated in the analysis of repeats transcription. B.N. participated in coordination of the project. N.M. conceived and designed FISH experiments, analyzed and

interpreted the results and participated in writing the manuscript. P.W. participated in the project coordination and funding acquisition. E.G.J.D. conceived and designed the research, coordinated the project, performed orthology and comparative analysis in nematode genomes, performed synteny and duplications analysis, analyzed and interpreted the ensemble of data and wrote the paper with main contributions from A-P.Z.M., E.D-S., N.M., K.R-S. and G.D.K.

## Competing interests

The authors declare no competing interests.
