## [Peer Review File · Nature Communications]

Unzipped genome assemblies of polyploid root-knot nematodes reveal unusual and clade-specific telomeric repeatsREVIEWER COMMENTS

Reviewer #1 (Remarks to the Author):

In this manuscript the authors used PacBio long-read sequencing technology to assemble high quality genomes for three species of parasitic apomictic polyploid root-knot nematodes from the genus *Meloidogyne*. The complex evolution history of these species, being hybrid polyploids has previously made it challenging to generate genome assemblies. The genomes presented here are much more contiguous than previously published versions and appear to be “unzipped” – as in the different subgenomes are assembled separately. The authors report that the typical telomeric components for nematodes are absent in these genomes, including key proteins (e.g., telomerase) and specific repeats. A novel composite repeat with three different motifs is described in these species. Interestingly, all three species share one of the motifs but differ in the two. These repeats were located at the ends of contigs from the genome assemblies, and almost exclusively on a single end. FISH experiments in *Meloidogyne incognita* confirm that the repeats are telomeric and located on single ends of chromosomes. Additionally, the authors identified transcripts from short-read and Iso-seq transcriptomes containing these telomeric repeats in *M. incognita*. Unlike some species that use transposable elements (TEs) for telomere lengthening the authors report that these repeats are not TEs or associated with TEs. Instead, and supported by the transcriptome evidence, the authors propose that the maintenance of telomeres in *Meloidogyne* may be alternative lengthening of telomere (ALT) mechanism.

Overall, the data generated appears to be high quality and the demonstration of a telomere distinct from other nematodes is strongly supported. The combination of genomic, transcriptomic, and FISH results make a clear case for a unique telomeric structure/mechanism at play. However, the “unzipped” nature of these assemblies is underutilized in a way that leaves some important questions unaddressed. For this reason, expanded on below, I think major revisions are warranted.

The complex nature of the genome evolution in this system is not really a part of the story. The context of these species being hybrid polyploids is never discussed. I am curious how these species have distinct telomere repeats when they have subgenomes with more recent common ancestry across species than within? Do subgenomes A and B in *Minc* look similar in terms of telomere content? How do subgenomes A and B compare across species? I think this should be answerable with unzipped genomes.

The comparative analysis between the three species could be stronger. It seems that *Minc* was decided as a focal species for this project as it's the only species with FISH and transcriptome analysis, but I didn't get a sense of why this species had more focus on it. Certainly, a figure like Supplemental Figure 7, which depicts the distribution of the different telomeric motifs at the ends of chromosomes for *Minc* should be generated for *Mjav* and *Mare*, too. The nature of the repeats in *Minc* vs. *Mjav* and *Mare* appears to be fairly different. The density plots of repeats and G4-quadruplex for *Mjav* and *Mare* indicate that the repeats are far less abundant than *Minc*. This observation is mentioned but not discussed. Could there be any differences in assembly quality affecting this finding? Ideally, FISH would be conducted for *Mjav* and *Mare*, too. I think the authors should at least explain in response how *Minc* was chosen as the only species analyzed via FISH. For the manuscript more discussion about the biological implications for the differences in repeat and G4-quadruplex densities is needed. For example, is chromosome stability influenced by these features?

Similarly, RNA-seq datasets exist for *Mjav* and *Mare* on SRA; are these data not suitable for the analyses performed on *Minc*? I think Iso-seq is lacking for the other two species. But unless there are technical reasons to not include *Mjav* and *Mare* transcriptomes, these data should be included for analysis. The possibility of TERRA would be much more compelling if these two other species, with different motifs, also expressed their repeats. Alternatively, if there is no evidence for transcription of the telomere repeats in *Mjava* or *Mare*, then the TERRA as a part of telomere maintenance is

questionable.

What are the evolutionary implications for losing telomerase and, presumably, not relying on substitutes (e.g., transposons)? Is there a possible link between noncanonical telomere maintenance and polyploidy in this system? Specifically, a recombination-based ALT pathway seen in some cancer cells is mentioned in the Discussion (Line 510) and cancer cells have notoriously wild genome structures. I think, in addition to the more immediate follow-up questions outlined in the discussion, some broader connections to genome evolution should be addressed. For example, the karyotypes in these species vary considerably but their apomictic reproductions may not be as affected much without chromosome pairing during meiosis.

Specific comments:

The ancestral state analysis is not present in the Methods. The supplemental figures refer to a reconstruction done with Mequite but this is not described in the main text. Furthermore, I don't see any text describing where the species tree came from. Is this a newly generated tree for this manuscript? If so, the methods for producing it should be included along with a figure that includes statistical support.

Generally, in the Genome Assembly section of the methods section, programs cited lack version numbers.

The phrase "unzipped" is undefined in this manuscript. I believe it is like phasing but should be defined in the context of polyploidy.

Line 308-309: It looks like (Supp Fig 7) Minc contig 291 has repeats at both ends, at least a composite repeat on one end and a motif 2 repeated on the other.

Line 339: typo Minc

Line 531-534: How likely is it that these distinct telomeres could be exploited for pest control? Especially, given that motifs 2 and 3 are much less conserved within the genomes of Mjav and Mare. Can the authors pose, even theoretically, how pest management could target telomeres and how that would be more efficient than current practices? This point is also made in the abstract, which would be appropriate if there's some greater plausibility described in the main text.

Table 3: I think these results refer only to the short-read RNA-seq. If so, that should be noted in the legend.

Figure 2 legend: The different shapes and stacked order of motif legends should be explained in the legend.

Sup Fig 7: I think e-value and p-value are mixed up or used interchangeably?

Reviewer #2 (Remarks to the Author):

The manuscript by Mota and colleagues concerns identification of novel telomeres in the plant parasitic nematode genus *Meloidogyne*. The data suggest that telomerase and TTAGGC telomere repeats have been lost multiple times during nematode evolution. This has been observed before for *Diploscapter*, which has a single, possibly circular chromosome. The authors found that long read sequencing of

Meloidogyne species failed to reveal the telomere sequence (TTAGGC)_n or the telomerase reverse transcriptase noted for the distantly related nematodes *Parascaris univalens* and *C. elegans*. The authors search short read genome sequences for many other nematodes and generally note the canonical telomere repeat TTAGGC and TERT, but failed to identify either in *Meloidogyne*, *Strongyloides* and *Trichinellidae* and *Diplogastridae* genera. Lack of an obvious TERT reverse transcriptase in *Meloidogyne* and possibly in the other genera suggests that ALT might maintain their telomeres.

The authors use long-read sequencing of *Meloidogyne* species and identify long ~80nt tandem repeats that are G-rich on one strand at one end of each contig (segment of genomic DNA), rarely at two ends. This might suggest that one chromosome end has a telomere and the other does not. Or else might suggest that contigs are often composed of part of a chromosome that is interrupted by a long segment of uncharacterized repetitive satellite DNA that cannot be spanned.

The authors perform DNA FISH to confirm that their repeats are at telomeres, but the metaphase spread shown does not clearly show that the FISH signal is at ends of DAPI-positive chromosomes. Much of the signal is away from chromosome ends and many of the FISH foci appear as large or larger than whole chromosomes, which might make it difficult to determine where the sequences are. The control FISH signal is very large in the prophase nucleus in Fig. 5, and the putative FISH telomere signal in Fig. 5 is not clearly at telomeres. Overall, the FISH data do not strongly support the premise that the identified repeats are at telomeres. The authors suggest that only one telomere per chromosome is labeled by FISH, but this is not clear. Furthermore, it is not consistent with hundreds of contigs but only ~50 chromosomes. It is reasonable to expect that all telomeres will be similar in structure. It is not clear to me how the authors can conclude that their telomere repeats are at a single chromosome end.

The authors identify RNA created from the putative telomere repeats, using short read and long read RNA seq data sets. 14 short read transcripts are identified. 22 isoseq reads were defined, one of which overlapped with a protein-coding gene. It is possible that these are transcripts from the authors' repeats. However, it should be noted that TERRA is a very rare RNA, typically only detectable by RT-PCR and almost impossible to detect by RNA seq. That said, the RNA seq data that the authors present may be accurate, but it does not mean that the repeat sequences are telomeres.

Overall, the authors have generated an interesting hypothesis from long-read sequencing data, but they do not back this up with convincing DNA FISH data.

points for authors:

- the authors suggest that *trt-1* is missing from *Strongyloidea* and *Trichinellidae*, but their method of genome or proteome interrogation for *trt-1* based on Panther analysis may be inaccurate. Perhaps use a *trt-1* protein in a species closely related to *Strongyloidea* and *Trichinellidae* for the comparison.

- The authors find a single copy of the telomere repeats at an internal segment of the genome. It has previously been reported that the TALT mechanism (Template of ALT) can create telomeres that are composed of tandem repeats of a segment of unique sequence genomic DNA next to an ITS tract. How likely is it that this copy is the ancestral source of DNA that was amplified to create *Meloidogyne* telomeres when telomerase was lost? Could this internal repeat be the source of all telomere repeats?

- 'Thus, excluding those contigs, the repeat arrays were present at the beginning of 25 contigs and at the end of 26 other contigs but never present at both extremities. We noted that, when present at the beginning of contigs, the consensus sequence was repeated on the positive strand (22/25 contigs) while when present at the end, it was repeated on the negative strand in reverse complement.' What is the significance of this orientation relative to the beginning or end of a contig? This might have to do with how the long-read sequencing was done and how the contigs were assembled.

- The authors identify repeated consensus sequences at the ends of many contigs that are composed of 3 multiple motifs, one or two of which are G-rich depending on the species. Interestingly, the G-rich

motifs may be oriented 5' to 3' towards the ends of the contigs, consistent with canonical telomere G-strands. If so, this would help support their case for the repeats being telomeres.

- 'Overall, the composite terminal repeats of Minc, Mjav and Mare display G-rich stretches and are predicted to form G-quadruplexes. Although these features are reminiscent of known telomeres in eukaryotes, it must be confirmed that these repeats are actually present at chromosome extremities and not the result of assembly artifacts.' I am not sure about the logic here. These repeats are present at the ends of 57 contigs. If their orientation is 5' to 3' towards contig termini bases on the G-rich strand, this is just like telomeres. The G-rich motifs are good candidates for G quadruplex formation. However, the fact that 12/57 repeat arrays overlap with a gene suggests that these 12 repeat arrays may not be contigs but may interrupt genes within a chromosome. It would be helpful to have cartoon models for the 12 genes adjacent to telomere repeats.

- 'In *Drosophila* and other Diptera insects that lack telomerase, retro-transposons have replaced simple telomeric repeats. Furthermore, previous analyses have suggested transposable elements are active in Minc37. Therefore, we investigated whether the composite terminal repeats identified in mitotic parthenogenetic *Meloidogyne* genomes could be related to transposable elements. We used the EDTA software³⁸ to predict and annotate transposable and other repetitive elements in the Minc, Mjav and Mare genomes (Supplementary Table 7). We checked whether some repetitive elements were predicted in the regions containing the *Meloidogyne* composite repeats. In Minc, EDTA predicted two repetitive sequences in these regions, one described as unclassified repetitive regions and the second one annotated as a putative CACTA TIR transposon (MITE/DTC code). Motifs 3 and 2 of the Minc composite terminal repeat were present in the EDTA sequence TE00000264, unassigned to a known (retro)transposon family, while motif 1 matched the repetitive sequence annotated as a CACTA TIR DNA transposon (TE00000823). We found this DNA transposon annotation surprising since transposable elements described as playing the role of telomeres are rather retro-transposons. We further investigated this case and no evidence for a transposase or any of the protein-coding genes usually found in CACTA DNA transposon could be identified in this predicted repeat. Furthermore, searching this annotated repeat (TE00000823) against the Repbase²⁸ database³⁹ returned no significant hit. Hence, this assignment as a CACTA DNA transposon is most likely an annotation error from EDTA.'

This is a long paragraph whose conclusion suggests an annotation error. Perhaps much of this information could be moved to the supplemental data.

The concluding paragraph seems germane: 'No evidence for a transposase or any other protein-coding gene could also be found in the Minc, Mjav or Mare composite terminal repeats themselves, and these sequences returned no hit against Repbase²⁸. Therefore, no evidence supports any of the identified terminal repeats could be related to a known transposable element.'

- the G-quadruplex region is similarly wordy and could be compressed into the following 'In Minc, we could observe a clear enrichment of segments forming G4-quadruplexes in the genomic regions corresponding to the terminal repeat arrays, with the rest of the genome being otherwise poor in G4-forming segments (Figure 3, Supplementary Fig. 9). Similar results were observed in the Mjav and Mare genomes'

- the G-quadruplex section ends 'Overall, the composite terminal repeats of Minc, Mjav and Mare display G-rich stretches and are predicted to form G-quadruplexes. Although these features are reminiscent of known telomeres in eukaryotes, it must be confirmed that these repeats are actually present at chromosome extremities and not the result of assembly artifacts.' Are the authors saying that the results of their G-quadruplex analysis are uncertain because they are not certain about the identity of telomeres? Or are the authors trying to create a transition to a DNA FISH section?

- 'Indeed, > 50 evaluated nuclei showed the repeat form arrays at telomeric and subtelomeric regions

of almost all chromosomes (Figure 4). We repeatedly observed only one or two chromosomes without signal.'

Figure 4 shows many chromosomes without signal (to my eyes), in contrast to the above statements. And a lot of FISH signals appear away from strong segments of DAPI DNA. If so, this means that there is a segment of uncondensed DNA that separates metaphase chromosomes from their telomeres. Or else it might mean that some or all of the FISH signal is an artefact.

- For the FISH data, the authors show a single metaphase. The signal intensity of the green FISH stain was often as large as or larger than whole metaphase chromosomes as assessed by DAPI. There is no metaphase control with single copy genes to show that their FISH assay is working.

- Although there is a prophase example in Fig. 5 with a single copy FISH control, it is very hard to say whether or not the putative telomere FISH signal in prophase is at telomeres. In addition, both control and telomere spots in Fig. 5 are mostly very large. The authors argue that the number of spots corresponds to the number of chromosomes (one telomere per chromosome labeled), but the number of FISH spots may not be a good estimate of telomere number. The authors should quantify the number of FISH spots per prophase nucleus and compare this with chromosome number or estimated telomere number. With enough nuclei quantified, their case for the telomere spots being telomeres might strengthen.

- 'The number of contigs bearing the composite repeats in the three species are consistent with these cytogenetics analyses and suggest our assemblies constitute a solid base to further scaffold the genomes at end-to-end chromosome scale resolution.' The authors suggest 45-47 chromosomes are expected per species and if they obtained 291-377 contigs based on long-read sequencing data, then this might translate into roughly 6 contigs per chromosome. If so, then this does not explain the presence of 'telomere motifs 1-3' at one end of each contig. Some nematode genomes are composed of large segments of tandem repeats (*Parascarus univalens*). Is it possible that the tandem repeat motifs identified by the authors are internal tandem repeat tracts? It is possible that the authors' contigs are at one end of each contig because each contig represents the end of a chromosome. The authors should quantify contigs with telomere repeat arrays at their ends and compare this with contig number that lack telomere repeat arrays at their ends and compare this with the expected telomere number to see if the repeat arrays might be telomeres.

- consistent with the above concern that these telomere repeats are found at one end of each contig and that there are many contigs per chromosome, the authors state in the Discussion 'Furthermore, unlike in *Drosophila*, these repeats are not retrotransposons and seem to be unrelated to any known transposable elements and more likely constitute satellite DNA.' While it is possible that these repeats are telomeres, how do the authors know that they are not internal segments of satellite DNA that nanopore sequencing cannot span?

Minor comments

- 'No evidence for a protein or domain interacting with single-stranded telomeric repeats could be identified in any *Meloidogyne* species'. Specify the shelterin protein POT1 here.

- 'CLK-2 is an ortholog of human TELO-2 telomere maintenance protein'. TEL2 protein.

- 'However, in 12 cases, one gene was predicted, usually starting upstream and overlapping the beginning of the repeat array.' No gene or chromosome is indicated for this data.

- G-loop is not a term used often in telomere biology. Perhaps T-loop?

Reviewer #3 (Remarks to the Author):

The manuscript by Mota and co-workers describes the results of nanopore sequencing of the genomes of the three major root knot nematodes of the genus *Meloidogyne*. Previous analysis of these genomes, in parts by the same lab, had improved the original genome sequence that had missed the triploid (polyploid) state of many *Meloidogyne* species. The current analysis reports a very laudable reduction in contig number down to the order of around 300 contigs. While this is a very high improvement from a contig perspective, CEGMA and BUSCO analysis indicates that the highly fragmented previous genomes had not missed many genes. Therefore, this manuscript focuses on what many nanopore genome sequencing projects do, namely looking at telomeres. Nearly half of the manuscript deals with an all-nematode comparative analysis of telomeres – repeat analysis, telomerase proteins and other related factors. Only in the last part of the manuscript, the authors come back to *Meloidogyne* and analyze the newly identified complex telomers and their localization through FISH.

This manuscript is premature in many parts and much of the analysis is incomplete. The following are the major concerns that must be addressed in order to make this manuscript a publishable unit.

1. While the devastating RKN of *Meloidogyne* are all parthenogenetic and polyploid, the temperate species *Meloidogyne hapla* is diploid, has a small genome and is clearly 'the model' of the genus. The authors should include *Meloidogyne hapla* in their genome analysis using nanopore sequencing. This would provide an internal control of their methodology. This should be an easy experiment and would be extremely helpful for the field.
2. *Meloidogyne hapla* is also excluded from the bioinformatic analysis, an unacceptable short coming for any evolutionary and phylogenetic analysis in this genus. One gets the impression that the authors want to hide even the existence of this species. This is scientifically flawed.
3. Why did the authors not use HiC analysis to bring the contig number even further down to chromosome number? Was that tried and it didn't work because of repeat structure? This is state-of-the-art in nematodes, and PacBio and HiC together result in chromosome number genomes throughout the phylum.
4. The finding of the presumptive terminal repeats in around 50-60 contigs coincides with chromosome counts and the FISH data are very interesting. However, finding these repeats on only one end of the chromosomes, results in important questions. First, can the authors rule out the idea that *Meloidogyne* has circular chromosomes? That would also explain why you do not need telomerases etc. Do you see your repeats in the middle of some of your long reads? If so, that would be some indication for this unconventional idea.
5. Second, if the chromosomes would indeed not be circular, what are the sequences at the other end of the chromosomes? The authors could do easily a FISH analysis with sequences of their other contig ends. Given they have a round 300 contigs (minus the ones that carry the repeats, leaves 250 contigs). Just checking 10-20 contig sequences by FISH should yield some chromosome ends to address the question: Is the second chromosome end (if existing and not circular), identical/related between chromosomes, or different? Are there any motifs that have gone undetected so far?

These are all major concerns that the authors must address to present a complete story that would come close to a publication in a major journal. In the current form, this third generation *Meloidogyne* sequencing is just incomplete.

Reviewer #1 (Remarks to the Author):

In this manuscript the authors used PacBio long-read sequencing technology to assemble high quality genomes for three species of parasitic apomictic polyploid root-knot nematodes from the genus *Meloidogyne*. The complex evolution history of these species, being hybrid polyploids has previously made it challenging to generate genome assemblies. The genomes presented here are much more contiguous than previously published versions and appear to be “unzipped” – as in the different subgenomes are assembled separately. The authors report that the typical telomeric components for nematodes are absent in these genomes, including key proteins (e.g., telomerase) and specific repeats. A novel composite repeat with three different motifs is described in these species. Interestingly, all three species share one of the motifs but differ in the two. These repeats were located at the ends of contigs from the genome assemblies, and almost exclusively on a single end. FISH experiments in *Meloidogyne incognita* confirm that the repeats are telomeric and located on single ends of chromosomes. Additionally, the authors identified transcripts from short-read and Iso-seq transcriptomes containing these telomeric repeats in *M. incognita*. Unlike some species that use transposable elements (TEs) for telomere lengthening the authors report that these repeats are not TEs or associated with TEs. Instead, and supported by the transcriptome evidence, the authors propose that the maintenance of telomeres in *Meloidogyne* may be alternative lengthening of telomere (ALT) mechanism.

Overall, the data generated appears to be high quality and the demonstration of a telomere distinct from other nematodes is strongly supported. The combination of genomic, transcriptomic, and FISH results make a clear case for a unique telomeric structure/mechanism at play. However, the “unzipped” nature of these assemblies is underutilized in a way that leaves some important questions unaddressed. For this reason, expanded on below, I think major revisions are warranted.

Question 1: The complex nature of the genome evolution in this system is not really a part of the story. The context of these species being hybrid polyploids is never discussed. I am curious how these species have distinct telomere repeats when they have subgenomes with more recent common ancestry across species than within? Do subgenomes A and B in *Minc* look similar in terms of telomere content? How do subgenomes A and B compare across species? I think this should be answerable with unzipped genomes.

Answer:

We fully agree with Reviewer 1 and have now placed the discovery of new telomeric repeats in the perspective of the allo-polyploid nature of the genomes. Indeed, previous analyses have shown that some subgenomes are more closely related between species (*Minc*, *Mjav* and *Mare*) than within species (Blanc-Mathieu et al. *PLoS Genetics* 2017, Lunt et al. *PeerJ* 2014, Jaron et al. *Journal of Heredity* 2021). Therefore, it might be expected that telomeric repeats would be more closely related among subgenomes between species than between subgenomes within a species.

We have now investigated this hypothesis in the light of the unzipped long read genome assemblies.

Using Smudgeplot analyses, we confirm that Minc more likely has an AAB structure while Mjav and Mare have a AABB structure (supplementary Figure1), consistent with previous results from the literature.

In this revised manuscript, we attempted to identify the A and B subgenomes in Minc, Mjav and Mare.

We first retrieved all the genomic blocks present in three copies in Minc (according to McScanX synteny results) and analyzed the distribution of the rate of synonymous mutations (Ks) between gene copies in these genomic blocks.

Consistent with the AA'B genome structure, we found a two-peak distribution with one relatively low Ks value corresponding to the relationship between A and A' and two relatively higher Ks values corresponding to the relationships between B and the two AA' as represented in the schematic figure below and (Supp. Figure 5):

Using these properties, whenever we observed two relatively higher Ks values and one relatively lower Ks value in the triplicated blocks, we could distinguish the B from the As contigs (A and A' being equidistant from B, it was not possible to further identify them).

Doing so, we unequivocally identified 113 A contigs and 43 B contigs in Minc, yielding identification of 29 A telomeres and 19 B telomeres, respectively.

To investigate A and B subgenomes in Mjav and Mare, we adapted the same strategy to tetraploid genomes. In this case, we selected all the genomic regions present in 4 copies and also studied the distribution of Ks values.

The most frequently observed relations between regions was that represented in the schematic figure of the quadruplet below:

M. incognita

M. javanica

M. arenaria

These Ks relationships were consistent with AA'BB' genomes with two relatively lower Ks values representing relationships between respectively A and A' and between B and B'; while four relatively higher Ks values represented the relationships between the two A and the two B subgenomes (Supp. Fig. 5).

Although this method allowed separating the quadruplets in two groups of two, this information alone was not sufficient to estimate which group represented A subgenomes and which one represented B subgenomes.

Therefore, we extended this intra-species Ks comparison to a between-species Ks-comparison. We performed a cross-species McScanX analysis, including the Minc, Mjav and Mare genomes and selected all the duplicated genome regions that were in 3 copies in Minc, 4 copies in Mjav and 4 copies in Mare and shared synteny between the 3 species. We calculated Ks values between all the intra and inter species relations and used these values to calculate a distance between the subgenomes and draw classification trees.

These trees confirmed that some subgenomes were more closely related between species than within species, consistent with previously reported hybridization events and more closely related ancestry within a subgenome between species than between subgenomes within species.

For each tree containing the three species, we assigned the Mjav and Mare subgenomes to either A or B according to their closest position relative to the previously assigned A and B Minc subgenomes in the trees.

This allowed unequivocally identifying:

- 64 A contigs and 56 B contigs in Mare including 15 A telomeres and 12 B telomeres.

- 58 A contigs and 62 B contigs including 11 A telomeres and 13 B telomeres in Mjav.

Then, we investigated the conservation of telomeric repeats within and across species taking into account whether they were present on A or B contigs. Because motif-1 is the only one conserved between the telomeric repeats of the three species, we used it as a basis for these comparisons. We retrieved all the occurrences of the conserved motif-1 in the three species, resulting in 2,660, 606 and 1,743 sequences in the Minc, Mjav and Mare genomes, respectively. We used CD-HIT to cluster motif-1 sequences at 99.9% identity with a minimal coverage threshold of 90% of the length of the Motif.

This resulted in 52 clusters ranging in decreasing size from 1,392 to 1 single sequence (Supplementary Table 7).

Analysis of the clusters revealed that 85.5, 86.2 and 85.5% of motif-1 occurrences, in Minc, Mare and Mjav, respectively, fell in clusters composed of both A and B contigs, indicating Motif1 is mostly indistinguishable between A and B subgenomes.

Extending the analysis to cross-species comparisons, we found that 4,288 motif-1 occurrences were in 14 multi-species clusters while only 771 were in 38 small species-specific clusters. Nine of the 14 multi-species clusters, covering 97.8% (4,192/4,288) of motif-1 occurrences, were composed of motifs coming both from A and B contigs, indicating again no separation between A and B repeats.

Overall, we can conclude that A and B subgenomes in Minc indeed look similar in terms of telomeric motif sequences. The same observation can be made within Mare and within Mjav. Furthermore, extending the analysis to cross-species comparisons, no evidence supports a separation between motif-1 from A contigs on one side and from B contigs on the other side.

Considering the whole repeat unit itself, as further detailed in the reply to the next question of Reviewer 1, there is also clear evidence that each species has a different repeat unit organization and that the Minc repeat is more different than those of Mare and Mjav. This is clear when comparing the telomeric units of Minc with those of Mare and Mjav (Figure 2). In addition to the sequence itself, a species-specific pattern in the organization of telomeric repeats can also be observed, characterized by the fact that the telomeric units in Minc are organized as dense tandem repeats, whereas the organization in Mare and Mjav indicates the existence of intermediate sequences between telomeric repeat units yielding lower density (Sup Figures 14-16)

Species-specific repeat units and clustering of motif-1 occurrences without grouping according to A or B contigs suggests that the telomeric repeats have evolved independently in each species after the hybridization events rather than being inherited from putative ancestors of the A and B subgenomes.

We can hypothesize that a telomeric variant has probably spread across all three subgenomes after each of the hybridization events in each species by concerted evolution. Concerted evolution is characteristic for tandemly repeated sequences. This is a two-level process in which changes in repeat units are homogenized throughout members of a repetitive family, and concomitantly fixed within a group of reproductively linked organisms. As a consequence of concerted evolution, tandemly repeated telomeric families became species-specific, regardless of subgenomes.

All the results about the identification of A and B contigs and telomeres, their comparison and the discussion have now been added in the revised manuscript.

Question 2: The comparative analysis between the three species could be stronger. It seems that Minc was decided as a focal species for this project as it's the only species with FISH and transcriptome analysis, but I didn't get a sense of why this species had more focus on it. Certainly, a figure like Supplemental Figure 7, which depicts the distribution of the different telomeric motifs at the ends of chromosomes for Minc should be generated for Mjav and Mare, too.

Indeed, *Meloidogyne incognita* is our most advanced "model" of polyploid root-knot nematodes. This species is the one we continuously rear in the lab and thus for which we have more material for further biological experiment. The other two species Mjav and Mare are only occasionally multiplied and with more difficulties.

Nonetheless, we fully agree that with all the available data, there is no reason to put more focus on Minc than for Mjav and Mare.

Therefore, we completely changed the whole part concerning the discovery of putative telomeric repeats in Mjav and Mare. Instead of starting from Minc repeats to identify Mjav and Mare contigs returning hits with this putative telomeric repeats, we now used the same *de novo* search strategy for each species independently and compared the results at the end. For each species we retrieved all the contig extremities and used two rounds of MEME and MAST search to identify enriched combinations of motifs. From these results we deduced a composite repeat in each species and searched it with BLAST against the whole respective genomes to study their distributions on the contigs.

Based on these results, we produced the same representation as the previous Supp. Fig. 7 for Mjav and Mare as well (now Supplementary Figures 8-10 and 14-16) .

In addition, we compared the nine identified motifs (3 motifs x 3 species) together to better characterize similarities and differences in the repeats themselves (New Figure 2). We used two approaches for that (i) direct comparison of all the MEME motifs using TomTom and construction of a dendrogram based on the e-values of the pairwise comparisons (ii) multiple alignment of all the consensus sequences of the motifs then construction of a maximum likelihood phylogenetic tree with IQ-tree. Both analyses converged in producing a classification tree showing three clades (New Figure 2).

In agreement with previous observations, motif-1 (M1) is highly conserved in all three species (first clade, with a rectangle symbol). A second clade (round symbol) consists of sequences that have a conserved portion of ~27bp in length. Among them, motif-2 of Mjav (MjM2) and motif-3 of Mare (MaM3) are very similar sequences, while motif-2 of Mare (MaM2) is a divergent sequence except for the ~27bp conserved part. Finally, the third clade (star symbol) consists of three more divergent sequences characterized by stretches of Gs.

In the context of the evolution of telomeric repeats, it seems that within the repeating unit, motif-1 was subject to selective constraint, while the other parts of the repeating unit (motifs-2 and -3) changed rapidly, creating a species-specific profiles of telomeric repeats (species-specific suites of symbols in new Figure 2).

The nature of the repeats in Minc vs. Mjav and Mare appears to be fairly different. The density plots of repeats and G4-quadruplex for Mjav and Mare indicate that the repeats are far less abundant than Minc. This observation is mentioned but not discussed. Could there be any differences in assembly quality affecting this finding?

Indeed, the inter-repeat distance is higher in Mjav and Mare than in Minc, resulting in a higher density of repeats for Minc (Supp. Figures 14-16). Similarly motifs-2 and 3 of Minc have conserved patterns of repeated stretches of Gs resulting in a higher density of predicted G4-forming regions. This situation is probably not due to differences in assembly qualities but most likely represents a biological reality. Indeed, we used the exact same methodology to assemble the three species (i) we selected ONT reads with a minimal Qscore of 12 (ii) we assembled the genomes with NECAT (iii) we polished the assemblies with two rounds of Racon then one round of Medaka (ONT) then one round of Hapo-G (Illumina). Furthermore, the KAT k-mer analyzes all show that the k-mers present in the Illumina reads have been well captured in the assembly suggesting very good per-base quality and good completeness of all the genomes. As a consequence, the N50 values, numbers of contigs and coverage are very similar between the three species. Therefore, these differences most probably reflect a biological reality rather than differences in genome assembly qualities (now also confirmed with FISH in the next point).

Ideally, FISH would be conducted for Mjav and Mare, too. I think the authors should at least explain in response how Minc was chosen as the only species analyzed via FISH. For the manuscript more discussion about the biological implications for the differences in repeat and G4-quadruplex densities is needed. For example, is chromosome stability influenced by these features?

As introduced in reply to previous points, *Meloidogyne incognita* is our main model species for polyploid root-knot nematodes and is routinely reared in our lab. Therefore, it is easier to obtain material to conduct biological experiments on this species and this is why we first used it for FISH analysis. During the revision process, we have deployed efforts to multiply Mjav and Mare and managed to perform DNA FISH analysis on these species too. Our recent analyses confirm telomeric positions of the identified repeats in Mjav and Mare as well and we have changed the manuscript and produced new figures accordingly (New Supplementary Figure 20, updated Figure 5).

Whether chromosome stability is influenced by the repeats and G4 densities is difficult to tell. In the three species, similar variations in terms of karyotypes between populations are observed: 41-46 chromosomes in Minc, 51-56 chromosomes in Mare and 42-48 chromosomes in Mjav (Triantaphyllou 1985). A recent analysis suggested that, at least in plants, telomeric repeat abundance reflects chromosome dynamics in telomeric region and is also a consequence of the history of breakages, fissions, fusions in a species (Radka Vozárová et al. The Plant Journal, 2022). In the *Meloidogyne* we observe different average number of repeats per arrays (47 in Minc, 10 in Mjav and 31 in Mare) and different average repeat array lengths between species (16kb for Minc, 10kb for Mjav, and 20kb for Mare). However, it is difficult to tell whether what we observe at the contig level represents a real difference in the chromosomes, particularly because repetitive regions are challenging to assemble. We think at this stage of our knowledge we can just hypothesize differences in repeat density might have consequences but this will need to be further investigated in the future. Differences in repeat density and their possible consequences have been added to the discussion.

Similarly, RNA-seq datasets exist for Mjav and Mare on SRA; are these data not suitable for the analyses performed on Minc? I think Iso-seq is lacking for the other two species. But unless there are technical reasons to not include Mjav and Mare transcriptomes, these data should be included for analysis. The possibility of TERRA would be much more compelling if these two other species, with different motifs, also expressed their repeats. Alternatively, if

there is no evidence for transcription of the telomere repeats in Mjava or Mare, then the TERRA as a part of telomere maintenance is questionable.

We completely agree and although ISO-seq data is lacking and Illumina transcriptome data are less abundant for Mjav and Mare, these are still suitable for investigation of the telomeric repeats and potential TERRA. Therefore, we searched in Mjav and Mare transcriptome data and exactly as for Minc, we found the presence of the three telomeric motifs in the transcriptome of each species. Hence, the same kind of evidence for a putative TERRA exists in Mjav and Mare as well and this has been added in the revised manuscript (results, updated Table3 and new supplementary tables 12-13).

What are the evolutionary implications for losing telomerase and, presumably, not relying on substitutes (e.g., transposons)? Is there a possible link between noncanonical telomere maintenance and polyploidy in this system? Specifically, a recombination-based ALT pathway seen in some cancer cells is mentioned in the Discussion (Line 510) and cancer cells have notoriously wild genome structures. I think, in addition to the more immediate follow-up questions outlined in the discussion, some broader connections to genome evolution should be addressed. For example, the karyotypes in these species vary considerably but their apomictic reproductions may not be as affected much without chromosome pairing during meiosis.

Although it is tempting to hypothesize a link between polyploidy, absence of meiosis and noncanonical telomere maintenance in the Meloidogyne genus, there does not seem to have a direct relation between these features. Indeed, *M. graminicola*, *M. chitwoodii* and *M. hapla* which are three diploid meiotic species, also lack the canonical TTAGGC_n or other simple repeats as well as a telomerase and homologs of the *C. elegans* telomere-associated proteins. Therefore, it seems that loss of the telomerase-based system and presence of the noncanonical telomere maintenance system is a feature of all the Meloidogyne species explored at the genome level so far, regardless of their mode of reproduction.

Besides, we completely agree that variations in karyotypes observed in the polyploid species are probably helped by the absence of homologous meiotic pairing of chromosomes. The holocentric chromosomes themselves might also favor karyotypic plasticity as even chromosome parts can be segregated during mitosis or modified meiosis with holocentromeres. However, whether the peculiar telomere system might be implicated in this karyotypic instability is unsure since the diploid species seem to have stable karyotypes.

We now discuss this point in the revised version of the paper and have added more discussion about genome evolution in these species in general.

Specific comments:

The ancestral state analysis is not present in the Methods. The supplemental figures refer to a reconstruction done with Mequite but this is not described in the main text. Furthermore, I don't see any text describing where the species tree came from. Is this a newly generated tree for this manuscripts? If so, the methods for producing it should be included along with a figure that includes statistical support.

Thanks for pointing this missing information. We have now included a whole new paragraph in the methods to explain how the reference nematode tree was drawn and how ancestral states were reconstructed.

Generally, in the Genome Assembly section of the methods section, programs cited lack version numbers.

The version for each program used to assemble and polish the genome has now been indicated in the material and methods section.

The phrase “unzipped” is undefined in this manuscript. I believe it is like phasing but should be defined in the context of polyploidy.

Yes, this is the same principle as phasing haplotypes in diploid genomes but transposed to polyploid genomes and subgenomes. We made this distinction to highlight the fact that what we are separating are not alleles and haplotypes but subgenomes that result from hybridization events. We have clarified this point in the introduction

Line 308-309: It looks like (Supp Fig 7) Minc contig 291 has repeats at both ends, at least a composite repeat on one end and a motif 2 repeated on the other.

Some of the smaller contigs like contig 291, 288, 282, 281, 270 and 258 are actually entirely made of the repeat and are thus completely repetitive (Supplementary Table 3). These short contigs most probably represent unassembled fragments of telomeres.

Line 339: typo Minc

Thanks, this has been corrected.

Line 531-534: How likely is it that these distinct telomeres could be exploited for pest control? Especially, given that motifs 2 and 3 are much less conserved within the genomes of Mjav and Mare. Can the authors pose, even theoretically, how pest management could target telomeres and how that would be more efficient than current practices? This point is also made in the abstract, which would be appropriate if there's some greater plausibility described in the main text.

We agree that at this stage, this remains speculative. The discovery of these very specific telomeric repeats opens many perspectives and how exactly they function, which are the proteins and RNAs interacting with them remain to be discovered. In theory, interfering with these telomeres should have dramatic consequences on the survival of the parasites. Furthermore, because the telomeric sequences are very specific to these allopolyploid root-knot nematodes, the risk of unintentional effect on other species should be low. Using guide RNA and CRISPR to specifically target and cut the highly conserved motif-1 might be a possibility in the future. Another possibility would be using RNAi to target transcripts corresponding to the proteins interacting with the telomeres. This necessitates prior identification of these proteins and would be more suitable if these are orphan proteins lacking homology in other species. We have modified this part of the discussion, accordingly.

Table 3: I think these results refer only to the short-read RNA-seq. If so, that should be noted in the legend.

Yes, exactly and this has now been indicated in the legend.

Figure 2 legend: The different shapes and stacked order of motif legends should be explained in the legend.

Figure 2 has completely changed and now represents a cross-species comparison of the similarity between the motifs represented as a classification tree. The legend has been updated accordingly.

Sup Fig 7: I think e-value and p-value are mixed up or used interchangeably?

Yes, indeed, thanks for pointing out this inconsistency, values returned by MEME on these graphs are p-values not e-values. The legend has been corrected accordingly.

Reviewer #2 (Remarks to the Author):

The manuscript by Mota and colleagues concerns identification of novel telomeres in the plant parasitic nematode genus *Meloidogyne*. The data suggest that telomerase and TTAGGC telomere repeats have been lost multiple times during nematode evolution. This has been observed before for *Diploscapter*, which has a single, possibly circular chromosome. The authors found that long read sequencing of *Meloidogyne* species failed to reveal the telomere sequence (TTAGGC)_n or the telomerase reverse transcriptase noted for the distantly related nematodes *Parascaris univalens* and *C. elegans*. The authors search short read genome sequences for many other nematodes and generally note the canonical telomere repeat TTAGGC and TERT, but failed to identify either in *Meloidogyne*, *Strongyloides* and *Trichinellidae* and *Diplogastridae* genera. Lack of an obvious TERT reverse transcriptase in *Meloidogyne* and possibly in the other genera suggests that ALT might maintain their telomeres.

The authors use long-read sequencing of *Meloidogyne* species and identify long ~80nt tandem repeats that are G-rich on one strand at one end of each contig (segment of genomic DNA), rarely at two ends. This might suggest that one chromosome end has a telomere and the other does not. Or else might suggest that contigs are often composed of part of a chromosome that is interrupted by a long segment of uncharacterized repetitive satellite DNA that cannot be spanned.

The authors perform DNA FISH to confirm that their repeats are at telomeres, but the metaphase spread shown does not clearly show that the FISH signal is at ends of DAPI-positive chromosomes. Much of the signal is away from chromosome ends and many of the FISH foci appear as large or larger than whole chromosomes, which might make it difficult to determine where the sequences are. The control FISH signal is very large in the prophase nucleus in Fig. 5, and the putative FISH telomere signal in Fig. 5 is not clearly at telomeres. Overall, the FISH data do not strongly support the premise that the identified repeats are at telomeres. The authors suggest that only one telomere per chromosome is labeled by FISH, but this is not clear. Furthermore, it is not consistent with hundreds of contigs but only ~50 chromosomes. It is reasonable to expect that all telomeres will be similar in structure. It is not clear to me how the authors can conclude that their telomere repeats are at a single chromosome end.

We thank the Reviewer for the valuable feedback on the visual representation of telomere localization with FISH experiments. Due to comments that the signals do not appear to be on the DAPI-stained chromosome we decided to include several additional materials that support our claims. Namely, we have provided:

- 1) Larger number of metaphase chromosomes where the DAPI-stained channel is shown in black and white. These additional images clearly show that strong telomere signals are

indeed at the chromosomal ends and coinciding with DAPI staining (see additional Figure 1 below and high-resolution version online <https://doi.org/10.57745/5SXZQH>).

2) Magnified regions of several chromosomes where a strong telomeric signal can be seen that can undoubtedly be attributed to the region of the chromosomes themselves when looking at the overlay on the DAPI channel shown in black and white (see additional Figure 2 below and high-resolution version online <https://doi.org/10.57745/5SXZQH>).

3) Animation of chromosomes alternately with telomere signals positioning shown through black and white DAPI representation. Video shows telomeres at the end of chromosomes which are weaker in DAPI staining but certainly occupied by the DNA (see new additional material - FISH_channels_video online <https://doi.org/10.57745/OWZLRN>).

4) Video animation of all z-stacks obtained with confocal microscopy which can help to understand the entire 3D telomere structure. Here it can be seen that the FISH signals are in the same z-axial plane as the DAPI stained chromatin (see new additional material FISH_3d_video <https://doi.org/10.57745/OWZLRN>).

5) Images of a larger number of interphases undoubtedly showing the specificity of signals on other nuclei phases taken from the same slides as shown metaphases (see additional Figure 3 below and high-resolution version online <https://doi.org/10.57745/5SXZQH>)

One explanation for weaker DAPI staining at the very end of chromosomes could be the specific structure of Meloidogyne chromosomes. Also, the FISH experiment itself could have a noticeable impact on chromosome morphology as it involves the step of chromosome denaturation at high temperature which can result in reduced preservation of nuclear morphology. However, the DAPI-stained chromosomes presented in black and white clearly show the telomeres are at the end of chromosomes. Furthermore, regarding the size of FISH foci, some of them can indeed seem to look as big as some of the smaller chromosomes. This is mostly a result of signal amplification procedure in FISH experiments where we use three layers of fluorophore conjugates and antibodies (fluorescein avidin D and biotinylated anti-avidin D) in order to additionally preclude telomere sequences is being present in a small number of copies at the other end. In that case, putative discrete signals could remain undetected but that showed not to be the case. Following the same rationale, we also additionally amplified the signal in silico in the images to make sure we do not miss even a weak signal. Taken together, this led to some signals appearing very strong compared to the chromosome size. We have added additional details for signal amplification in the FISH protocol in Methods section for better understanding of the procedure.

All additional figures are integrated within the new Supplementary Figure 18 and animated material is deposited in the data repository (Supplementary Figure 19).

Additional figure 1. Additional images (A-F) of telomeric chromosomal localization of the Minc composite repeat using fluorescence in situ hybridization in metaphase shown on many different chromosomes. FISH was performed as described in the Methods section. Chromosomes are counterstained with DAPI that is shown in black and white for better chromosomal end visualization. Arrows represent chromosomes with no visible telomere signal at the end.

Additional figure 2. Telomere localization of Minc repeats shown with enlarged chromosomes and accompanying FISH signals that occupy weaker stained ends of chromosomes (region marked with red arrows).

Additional figure 3. Localization of telomeric composite repeats on interphase nuclei showing the high specificity of FISH stainings together with very low background.

Concerning the difference between the total number of contigs (291 in *Minc*, 364 in *Mjav* and 377 in *Mare*) and the number of chromosomes (41-47 in *Minc*, 42-48 in *Mjav* and 51-56 in *Mare*), this is simply due to our assembly not yet being resolved at a chromosome-scale resolution. Thus, some chromosomes are represented by multiplied contigs. Some contigs are relatively short, including contigs entirely made of the telomeric repeat that more likely represent unassembled fragments of telomeres. However, we can note that the number of repeat arrays found on contig extremities (52 in *Minc*, 59 in *Mjav* and 57 in *Mare*) is close to the number of chromosomes in these species. This observation is consistent with the identified repeats being present at only one end of most chromosomes.

The authors identify RNA created from the putative telomere repeats, using short read and long read RNA seq data sets. 14 short read transcripts are identified. 22 isoseq reads were defined, one of which overlapped with a protein-coding gene. It is possible that these are transcripts from the authors' repeats.

However, it should be noted that TERRA is a very rare RNA, typically only detectable by RT-PCR and almost impossible to detect by RNA seq. That said, the RNA seq data that the authors present may be accurate, but it does not mean that the repeat sequences are telomeres.

We completely agree that finding transcripts that contain our candidate telomeric repeats is not sufficient alone to imply the sequence we identified is necessarily telomeric. The goal of this analysis was more to study whether similarly to other species, the candidate telomeric repeat was transcribed. Following one suggestion from reviewer 1, we also identified transcripts containing the candidate telomeric repeats in *Mjav* and *Mare* as well, reinforcing the idea that the candidate telomeric repeats are transcribed in the three species.

Overall, the authors have generated an interesting hypothesis from long-read sequencing data, but they do not back this up with convincing DNA FISH data.

We understand the concerns raised by the Reviewer regarding the FISH data. For this reason we provided several additional materials (explained in detail above) that speak in favor of specificity of experiments together with a much more detailed view of chromosomal ends (3D and animated material). Also, it should be noted that Meloidogyne species have very small and numerous chromosomes, so it is very difficult to obtain chromosome preparations at high resolution. Due to that, we often see partial sets of their chromosomes, or unclear borders

between chromosomes belonging to one or another set. For comparison, the *M. incognita* population analyzed in this work has about 46 chromosomes that are between 0.4 and 1.5 μm in size in metaphase, while *C. elegans* has only 5 pairs of much larger chromosomes (5 μm). Nevertheless, all newly added material shows specific FISH signals that occupy the very end of most chromosomes only on one side. As suggested by Reviewer 1, we also produced similar FISH results for *M. javanica* and *M. arenaria* reinforcing the finding that the repeats we have identified in the three species have specific telomeric localization.

points for authors:

- the authors suggest that trt-1 is missing from Strongyloididae and Trichinellidae, but their method of genome or proteome interrogation for trt-1 based on Panther analysis may be inaccurate. Perhaps use a trt-1 protein in a species closely related to Strongyloididae and Trichinellidae for the comparison.

We have actually performed a comprehensive and extensive search for trt-1 and all the known telomere-associated proteins from *C. elegans* in a total of 69 nematode genomes and proteomes (now including *Meloidogyne hapla* following Reviewer 3 comments). As explained in the methods, this thorough search was not limited to a search for conserved domains but also included a comprehensive all-against-all comparative analysis of the predicted proteins from the 69 nematodes using Othofinder. Orthofinder performs an all-against-all comparison of all the proteins of each species against all the proteins of all the other species. Based on reciprocal better hits relations, OrthoFinder then constructs OrthoGroups.

Therefore, all the proteins of Strongyloididae and Trichinellidae have already been compared with telomerases and all the other proteins from the most closely related species included in our analysis. No reciprocal best hits with telomerase proteins of any species including the most closely related ones could be retrieved in species from these genera.

However, because this could be due to the genes being not predicted in these genomes, we also performed a tBLASTn search using *C. elegans* and *Ascaris suum* proteins (whenever available) against the genomes.

Only when there was lack of evidence from the three approaches (domain-based, OthoFinder, tBLASTn) we concluded the gene was most likely absent.

Given no evidence for any of the three methods is present for the Strongyloididae and Trichinellidae, we were quite confident the genes are actually absent from these species too. This also makes sense with the absence of canonical TTAGGCn telomeric repeats either in the genomes or raw sequencing reads for these species (Figure1).

Nevertheless, as an additional confirmation, we performed a tBLASTn against the Strongyloididae and Trichinellidae genomes using the Trt-1 homologs from the most closely related species available.

For the Strongyloididae we used protein L596_019393 from *Steinernema carpocapsae* which is the closest available outgroup, has a telomerase reverse transcriptase domain and is in the same OrthoGroup than the biochemically confirmed telomerase of *Ascaris suum*. The search with an e-value of $1\text{E-}3$ returned no hit in any of the Strongyloididae genomes further supporting absence of telomerase in this phylogenetic group.

For the Trichinellidae, finding a closely related species with a telomerase was more complicated. Indeed, no species within the whole Clade I had unequivocal evidence for a telomerase. Only *Trichuris suis* (Trichuridae in Clade I) had partial evidence with a protein (D918_0798) containing a discontinuous telomerase reverse transcriptase domain and which

is not in an OrthoGroup with any biochemically confirmed telomerase. Using this protein as a query for a tBLASTn search against Trichinellidae returned multiple hits. However all of them were restricted to the 100 first amino acids of D918_0798, a region that does not contain the telomerase domain. Therefore, no further evidence was found in Trichinellidae as well.

- The authors find a single copy of the telomere repeats at an internal segment of the genome. It has previously been reported that the TALT mechanism (Template of ALT) can create telomeres that are composed of tandem repeats of a segment of unique sequence genomic DNA next to an ITS tract. How likely is it that this copy is the ancestral source of DNA that was amplified to create Meloidogyne telomeres when telomerase was lost? Could this internal repeat be the source of all telomere repeats?

This is a very interesting suggestion and is indeed tempting to speculate this repeat unit could be the ancestral source. Unfortunately, this unique full-length repeat is at the very beginning contig #248 from position 450 to 704 while the whole contig is ca. 50kb long. Therefore, we cannot really consider this repeat unit is internal and there is no additional supporting evidence suggesting it could represent the ancestral template of origin.

- 'Thus, excluding those contigs, the repeat arrays were present at the beginning of 25 contigs and at the end of 26 other contigs but never present at both extremities. We noted that, when present at the beginning of contigs, the consensus sequence was repeated on the positive strand (22/25 contigs) while when present at the end, it was repeated on the negative strand in reverse complement.'

What is the significance of this orientation relative to the beginning or end of a contig? This might have to do with how the long-read sequencing was done and how the contigs were assembled.

Oxford Nanopore sequencing technology directly reads one strand of DNA molecules one by one without amplification and therefore preserves the orientation on the sequenced molecule. Because our three genomes have been assembled from ONT reads, the orientation is also preserved on the contigs. Consequently, the contigs are supposed to be entirely either on the direct (+) orientation or reverse complement (-) orientation. The fact that we find the repeats either in the + orientation at the beginning or the - orientation at the end of the contigs mean the repeats are orientated and do not form palindrome or inverted repeats. However, it does not mean the repeats are either all at the beginnings or all at the ends of the chromosomes. The fact that they are orientated is reminiscent of all the other known telomeric sequences reported so far.

- The authors identify repeated consensus sequences at the ends of many contigs that are composed of 3 multiple motifs, one or two of which are G-rich depending on the species. Interestingly, the G-rich motifs may be oriented 5' to 3' towards the ends of the contigs, consistent with canonical telomere G-strands. If so, this would help support their case for the repeats being telomeres.

Yes, indeed, in each species the composite repeat consensus sequence is composed of one motif (motif-1) which is highly conserved within and between species and two others (motif-2 and motif-3) that are G-rich, with those of *Minc* being particularly G-rich. We completely agree with reviewer2 that it would be very interesting to determine whether as for the other species, the G-rich strand is the one orientated 5' → 3' towards the end of the contigs and thus chromosomes. We tried to infer this from transcriptomic data but because available RNA-seq data are not oriented we find both G-rich and their C-rich reverse complements in the

transcripts. Looking at ISO-seq data (only available for *Minc*), we found three long transcripts containing the motifs and with a Poly-A tail. These transcripts were relatively C-rich suggesting they have been transcribed from a G-rich strand. However, we think more evidence is needed to strongly conclude which strand is G-rich.

- 'Overall, the composite terminal repeats of *Minc*, *Mjav* and *Mare* display G-rich stretches and are predicted to form G-quadruplexes. Although these features are reminiscent of known telomeres in eukaryotes, it must be confirmed that these repeats are actually present at chromosome extremities and not the result of assembly artifacts.'

I am not sure about the logic here.

What we meant here is that even though these repeats share several features with known telomeres (they are stranded, G-rich, predicted to form G4-quadruplex and are transcribed), the fact that we find them at the extremities of contigs alone is not sufficient to conclude they actually constitute telomeric DNA. To eliminate this risk this would result from a genome assembly artifact and confirm telomeric localisation, performing FISH experiment was important and necessary. Indeed, long repetitive sequences can cause the assembly to stop at these repeats if not enough long reads pass through the repeats. Therefore, to make sure this was not the case, and that the repeat we identified had terminal localization just like telomeres, DNA FISH experiments were required.

We have now reorganized the manuscript to describe all the bioinformatics evidence suggesting the repeats are telomeric DNA (including the transcriptomic analysis), and then make a transition with the FISH experiments confirming telomeric localization. This transition sentence is now rephrased and moved at the beginning of the FISH section.

These repeats are present at the ends of 57 contigs. If their orientation is 5' to 3' towards contig termini bases on the G-rich strand, this is just like telomeres. The G-rich motifs are good candidates for G quadruplex formation. However, the fact that 12/57 repeat arrays overlap with a gene suggests that these 12 repeat arrays may not be contigs but may interrupt genes within a chromosome. It would be helpful to have cartoon models for the 12 genes adjacent to telomere repeats.

Indeed, 57 contigs possess arrays of the *Minc* repeat including 5 that are fully repetitive and thus 52 where the array is either at the beginning or at the end. In 12 cases, one single predicted gene overlaps with the repeat array. Considering FISH results and the fact that fluorescence signal has never been observed within or in the middle of a chromosome in any of the slides, it is very unlikely that these repeats are localized internally in a chromosome and interrupt one gene. However, we still further investigated these genes. Interestingly these 12 genes have only been predicted in *Minc*, *Mjav*, *Mare* and *Miuci* and none of them has homolog in any other nematode in the OrthoFinder analysis. It should be noted that these four species have been annotated with the same pipeline, with the same parameters and using the same transcriptomic data as a source of evidence. Furthermore, we noted that none of them was strongly supported by transcriptome data and most have very short coding sequences. Therefore, one possibility is that these genes result from overpredictions of annotation software and might not represent real genes.

We provide a representation of the 12 genes in JBrowse in the supplementary figure available online at <https://doi.org/10.57745/ECPWHV>

- 'In *Drosophila* and other Diptera insects that lack telomerase, retro-transposons have replaced simple telomeric repeats. Furthermore, previous analyses have suggested transposable elements are active in Minc37. Therefore, we investigated whether the composite terminal repeats identified in mitotic parthenogenetic *Meloidogyne* genomes could be related to transposable elements. We used the EDTA software³⁸ to predict and annotate transposable and other repetitive elements in the Minc, Mjav and Mare genomes (Supplementary Table 7). We checked whether some repetitive elements were predicted in the regions containing the *Meloidogyne* composite repeats. In Minc, EDTA predicted two repetitive sequences in these regions, one described as unclassified repetitive regions and the second one annotated as a putative CACTA TIR transposon (MITE/DTC code). Motifs 3 and 2 of the Minc composite terminal repeat were present in the EDTA sequence TE00000264, unassigned to a known (retro)transposon family, while motif 1 matched the repetitive sequence annotated as a CACTA TIR DNA transposon (TE00000823). We found this DNA transposon annotation surprising since transposable elements described as playing the role of telomeres are rather retro-transposons. We further investigated this case and no evidence for a transposase or any of the protein-coding genes usually found in CACTA DNA transposon could be identified in this predicted repeat. Furthermore, searching this annotated repeat (TE00000823) against the Repbase²⁸ database³⁹ returned no significant hit. Hence, this assignment as a CACTA DNA transposon is most likely an annotation error from EDTA.'

This is a long paragraph whose conclusion suggests an annotation error. Perhaps much of this information could be moved to the supplemental data.

The concluding paragraph seems germane: 'No evidence for a transposase or any other protein-coding gene could also be found in the Minc, Mjav or Mare composite terminal repeats themselves, and these sequences returned no hit against Repbase²⁸. Therefore, no evidence supports any of the identified terminal repeats could be related to a known transposable element.'

We have rephrased and reduced the whole section about the lack of evidence for relation to either DNA or retro-transposons. We think it is still important to mention this result in the main results because, in the absence of telomerase and canonical telomeric repeat, the most evident alternative hypothesis is that retro-transposons might replace the simple telomeric repeats such as in *Drosophila* and several other dipteran insects. In the *Meloidogyne*, this does not seem to be the case, suggesting ALT mechanism for telomeres.

- the G-quadruplex region is similarly wordy and could be compressed into the following 'In Minc, we could observe a clear enrichment of segments forming G4-quadruplexes in the genomic regions corresponding to the terminal repeat arrays, with the rest of the genome being otherwise poor in G4-forming segments (Figure 3, Supplementary Fig. 9). Similar results were observed in the Mjav and Mare genomes'

We rephrased and reduced this section as suggested.

- the G-quadruplex section ends 'Overall, the composite terminal repeats of Minc, Mjav and Mare display G-rich stretches and are predicted to form G-quadruplexes. Although these features are reminiscent of known telomeres in eukaryotes, it must be confirmed that these repeats are actually present at chromosome extremities and not the result of assembly artifacts.' Are the authors saying that the results of their G-quadruplex analysis are uncertain because they are not certain about the identity of telomeres? Or are the authors trying to create a transition to a DNA FISH section?

Yes, exactly, here we wanted to make a transition to the DNA FISH section. The rationale was to explain that despite several bioinformatics features reminiscent of known telomeric DNA were found, it was, in our opinion, essential to validate the localization of the sequences on the chromosomes using DNA FISH experiments. As indicated previously, we have now reorganized the manuscript to describe all the bioinformatics evidence suggesting the repeats are telomeric DNA (including the transcriptomic analysis), and then make a transition with the FISH experiments confirming telomeric localization. This transition sentence is now rephrased and moved at the beginning of the FISH section.

- 'Indeed, > 50 evaluated nuclei showed the repeat form arrays at telomeric and subtelomeric regions of almost all chromosomes (Figure 4). We repeatedly observed only one or two chromosomes without signal.' Figure 4 shows many chromosomes without signal (to my eyes), in contrast to the above statements. And a lot of FISH signals appear away from strong segments of DAPI DNA. If so, this means that there is a segment of uncondensed DNA that separates metaphase chromosomes from their telomeres. Or else it might mean that some or all of the FISH signal is an artefact.

On the first instance when looking at the metaphase shown in Figure 4 some chromosomes can look as if they have no signal. For that reason, we performed manual counting of chromosomes and signals that is now shown in additional Figure 4 below and in the supplementary material online (<https://doi.org/10.57745/5SXZQH>). There are 47 chromosomes on this metaphase plate, and this is consistent with the number of chromosomes previously reported in *M. incognita*. For the FISH signals, we counted 39 of them. There are two chromosomes without the signal (as already described in manuscript) and some of the telomere signals certainly overlap (and we only counted ones that we are sure belong to separate chromosomes) so the actual number is somewhere in between 40-45. Having everything in mind, together with counting performed on additional nuclei (see Figure 5 in answer below) we believe that our statement that the majority of chromosomes have a telomeric signal best describes the true situation we observe.

The FISH signals are indeed positioned at the very end of chromosomes with weaker DAPI staining probably due to specific telomere structure or denaturation step in the FISH procedure which has been added to Materials section. This is explained in detail within 2) point and additional Figure 2 above. Specificity of the FISH experiment is reinforced with new material (3D video animation (<https://doi.org/10.57745/OWZLRN>), DAPI shown in black and white) and additional figures (additional figures 1-5).

Additional figure 4. Quantification of chromosome (A) and telomere (B) number performed on Figure 4 from main manuscript text.

- For the FISH data, the authors show a single metaphase. The signal intensity of the green FISH stain was often as large as or larger than whole metaphase chromosomes as assessed by DAPI. There is no metaphase control with single copy genes to show that their FISH assay is working.

We agree with the Reviewer that it is difficult to assess the FISH data based on only one metaphase. For that reason, we have now shown many metaphases from different nuclei (additional Figure 1 above) which also supports the reproducibility of FISH results. Signal intensity and the size of the telomere signal is mostly discussed and explained within the previous answers. The strong signals are the result of signal amplification (added to Material section) that also allow excluding the hypothesis that the same telomere sequences are present in a small number of copies at the other end. If that was the case, putative discrete signals could have remained undetected in the absence of signal amplification. For this reason, we additionally amplified the signal *in silico* using the visualization software which allowed confirming the absence of signal on the other ends.

In order to demonstrate the specificity of the FISH experiments, we have added several supporting data including reproducible stainings on more metaphases (additional Figure 1), specific staining on interphase/prometaphase nuclei (Figure 5 below) and signal foci positioning along z-stack at the chromosomal ends (additional FISH_3d_video <https://doi.org/10.57745/OWZLRN>) as the most prominent ones.

- Although there is a prophase example in Fig. 5 with a single copy FISH control, it is very hard to say whether or not the putative telomere FISH signal in prophase is at telomeres. In addition, both control and telomere spots in Fig. 5 are mostly very large. The authors argue that the number of spots corresponds to the number of chromosomes (one telomere per chromosome labeled), but the number of FISH spots may not be a good estimate of telomere number. The authors should quantify the number of FISH spots per prophase nucleus and

compare this with chromosome number or estimated telomere number. With enough nuclei quantified, their case for the telomere spots being telomeres might strengthen.

As suggested by the Reviewer, we performed the quantification on telomere FISH images and included the signal spot analysis on several interphase or prometaphase nuclei. These phases are ones that are the most commonly found on microscopic preparation of *M. incognita*, unlike full metaphase chromosome sets which are very rarely seen and hard to obtain. This offered us an opportunity to carry out automatic thresholding and counting of signals as presented in additional Figure 5 below. We obtained consistent results with the number of spots in between 38 and 42 which is in line with the other shown data (Figure 4) and drawn conclusions about most chromosomes having the newly defined telomere repeats.

Additional figure 5. Quantification of telomere FISH signals on interphase and prophase nuclei (A-E) of *M. incognita*. Spot analysis was done on telomere (green) channel that was automatically thresholded and spots were counted based on their mutual separation. Panels are shown with merged channels, DAPI shown in black and white, FISH signals and ImageJ quantification respectively. Total number of telomere signals is given in the top right box on last panel for each analyzed nucleus.

- ‘The number of contigs bearing the composite repeats in the three species are consistent with these cytogenetics analyses and suggest our assemblies constitute a solid base to further scaffold the genomes at end-to-end chromosome scale resolution.’ The authors suggest 45-47

chromosomes are expected per species and if they obtained 291-377 contigs based on long-read sequencing data, then this might translate into roughly 6 contigs per chromosome. If so, then this does not explain the presence of 'telomere motifs 1-3' at one end of each contig. Some nematode genomes are composed of large segments of tandem repeats (*Parascarus univalens*). Is it possible that the tandem repeat motifs identified by the authors are internal tandem repeat tracts?

As explained in the response to previous remarks from reviewer 2, it is very unlikely that the repeats we have identified form internal arrays within the chromosomes. Indeed, we never observed a fluorescence signal in the middle of a chromosome or even internally within a chromosome 'arm' despite the computational amplification of the signal. In *Minc*, we repeatedly observed 38-42 spots corresponding to the repeat arrays. In metaphase, we counted 47 chromosomes, implying that most (80.5 - 89.4%) of the chromosomes have the telomeric repeat. We also provide multiple images showing evidence that the repeat has a terminal localization on the chromosomes and is found at only one extremity. Therefore, we can assume that most of the chromosomes possess the repeat and only at one end. Although this does not completely exclude some chromosomes that might have the repeat at both ends or internally, these cases must be at least rare enough to be never observed.

Regarding the number of contigs (291-377) compared to the expected number of chromosomes (41-47 in *Minc*, 42-48 in *Mjav* and 51-56 in *Mare*), indeed, this corresponds to 6-7 contigs / chromosomes on average by doing a simple division. However, the longest contigs that are 6-8Mb long probably already represent full chromosomes and as further explained in the next point, the number of repeat arrays found is more compatible with most chromosomes having an array at one single end than two and this is consistent with FISH validations of repeat localization.

It is possible that the authors' contigs are at one end of each contig because each contig represents the end of a chromosome. The authors should quantify contigs with telomere repeat arrays at their ends and compare this with contig number that lack telomere repeat arrays at their ends and compare this with the expected telomere number to see if the repeat arrays might be telomeres.

In this population of *Minc*, we counted 47 chromosomes in metaphase. This is consistent with a previous count (46) on another population (Despot-Slade et al. 2021, *Mol Biol Evol*). A massive cytogenetics analysis of root-knot nematodes by A.C Triantaphyllou in 1985 counted 41-46 chromosomes in 215 populations of *Minc* from 64 countries, which is also consistent with our counts. Therefore, we can be confident 47 is close to the exact number of chromosomes in the *Minc* population we have sequenced. If *Minc* chromosomes had the repeat at both ends, we would expect approximately 94 (2x47) repeat arrays on contigs and several contigs with the repeat at both ends. However, in the genome assembly, we identified 52 contigs with the repeat forming arrays and they were either at the beginning or at the end. This observation is more consistent with most chromosomes having the repeat at one single end and possibly a few unobserved ones with telomeres at both ends. Indeed, the number of arrays is close to the observed number of chromosomes and not close to twice the number of chromosomes. Even if we consider the unlikely hypothesis that the 5 shorter fully repetitive contigs would possibly represent chromosome ends unassembled with the rest of a chromosome and not extending previous arrays, the final number of arrays (57) would still be far from the expected number under the 'both ends' hypothesis (94). Furthermore, the FISH analysis supports the fact that most chromosomes have the repeat at one single extremity.

In Mare, the most frequently observed chromosome numbers are 51-56 on 68 populations from 32 countries (Triantaphyllou 1985). In the genome assembly, we observed 57 arrays on 53 contigs, close to the number of chromosomes and not twice the number of chromosomes. Thus, we can hypothesize that in Mare too, most chromosomes most likely have the repeat array at only one end.

Finally, in Mjav, 42-48 chromosomes have been counted in 126 populations from 45 countries (Triantaphyllou 1985). In the genome assembly, we observed 59 repeat arrays on 57 contigs and no fully repetitive short contigs. This number is consistent with a majority of chromosomes having the repeat at one end and 8-12 having repeats at both ends. Interestingly, the Mjav genome assembly is the only one in which we observed contigs (2) with repeats at both ends and with correct +/- orientation. This further suggests several chromosomes might have the repeat at both ends in Mjav.

These important clarifications have been added to the discussion.

- consistent with the above concern that these telomere repeats are found at one end of each contig and that there are many contigs per chromosome, the authors state in the Discussion 'Furthermore, unlike in Drosophila, these repeats are not retrotransposons and seem to be unrelated to any known transposable elements and more likely constitute satellite DNA.' While it is possible that these repeats are telomeres, how do the authors know that they are not internal segments of satellite DNA that nanopore sequencing cannot span?

As previously shown as a response to the previous points raised by reviewer 2, we now provide additional evidence that the satDNA repeat we have identified in MInc has specific terminal location and on most chromosomes. We never observed an intra-chromosome fluorescence signal despite all the signal amplifications efforts deployed. Therefore, it is extremely unlikely that these repeats could represent unassembled internal repeats.

Minor comments

- 'No evidence for a protein or domain interacting with single-stranded telomeric repeats could be identified in any Meloidogyne species'. Specify the shelterin protein POT1 here.

No evidence for an homolog of POT1,2 or 3 or MRT-1 or even a ss-telomericDNA-binding domain could be identified. We have modified the sentence accordingly.

- 'CLK-2 is an ortholog of human TELO-2 telomere maintenance protein'. TEL2 protein. We have modified the sentence accordingly.

- 'However, in 12 cases, one gene was predicted, usually starting upstream and overlapping the beginning of the repeat array.' No gene or chromosome is indicated for this data.

As discussed previously, these 12 genes are short, lack ortholog in species other than the four polyploid apomictic Meloidogyne we have sequenced and annotated, and are only partially supported by transcriptome data. Therefore, whether they represent real genes or result from over-prediction from gene annotation is uncertain. Consequently, we have decided to remove this part as this information is too uncertain and not relevant to the rest of the findings. Instead we have added additional evidence for telomeric position of the repeats identified as well as confirmation in Mjav and Mare with FISH experiments.

- G-loop is not a term used often in telomere biology. Perhaps T-loop?

Co-formation of G-quadruplexes and R-loops (DNA/TERRA duplex) results in special structures, called G-loops, which are postulated to facilitate the ALT recombination process (Yang et al. 2021, G-quadruplexes mark alternative lengthening of telomeres. NAR Cancer). Here, we specifically refer to G-loop because the repeats we have identified are predicted to form G-quadruplexes and we have evidence from transcriptome data that TERRA probably exist in these species. Furthermore, in the absence of telomerase and because the repeats are not (retro)-transposons, we suspect alternative lengthening of telomeres via recombination such as described in ALT. We agree that this section needed to be clarified to better understand the arguments in favor of ALT. We have edited this part of the discussion accordingly.

Reviewer #3 (Remarks to the Author):

The manuscript by Mota and co-workers describes the results of nanopore sequencing of the genomes of the three major root knot nematodes of the genus *Meloidogyne*. Previous analysis of these genomes, in parts by the same lab, had improved the original genome sequence that had missed the triploid (polyploid) state of many *Meloidogyne* species. The current analysis reports a very laudable reduction in contig number down to the order of around 300 contigs. While this is a very high improvement from a contig perspective, CEGMA and BUSCO analysis indicates that the highly fragmented previous genomes had not missed many genes. Therefore, this manuscript focuses on what many nanopore genome sequencing projects do, namely looking at telomeres. Nearly half of the manuscript deals with an all-nematode comparative analysis of telomeres – repeat analysis, telomerase proteins and other related factors. Only in the last part of the manuscript, the authors come back to *Meloidogyne* and analyze the newly identified complex telomers and their localization through FISH.

We thank the reviewer for this remark but it seemed important to us to bring solid evidence that most of the telomere-related genes in addition to the canonical telomere repeat were missing in *Meloidogyne* genomes and extend this information to as many nematode genomes publicly available and assembled at sufficient contiguity level. There is of course a focus on *Meloidogyne* in this whole first part of the manuscript. The second part of the manuscript, which is bigger than the first part, focuses on the finding of new telomeric repeats in these *Meloidogyne* species.

This manuscript is premature in many parts and much of the analysis is incomplete. The following are the major concerns that must be addressed in order to make this manuscript a publishable unit.

1. While the devastating RKN of *Meloidogyne* are all parthenogenetic and polyploid, the temperate species *Meloidogyne hapla* is diploid, has a small genome and is clearly 'the model' of the genus. The authors should include *Meloidogyne hapla* in their genome analysis using nanopore sequencing. This would provide an internal control of their methodology. This should be an easy experiment and would be extremely helpful for the field.

2. *Meloidogyne hapla* is also excluded from the bioinformatic analysis, an unacceptable short coming for any evolutionary and phylogenetic analysis in this genus. One gets the impression that the authors want to hide even the existence of this species. This is scientifically flawed.

We completely agree with Reviewer 3 that including *Meloidogyne hapla* in the analysis would really bring valuable information because this is the most studied diploid sexual *Meloidogyne* species and the closest outgroup to the polyploid ones we studied here and with genome data already available.

Our intention was not at all to exclude and disregard *M. hapla* from the whole analysis and we are sorry this was the impression given by our paper.

So far, the only *M. hapla* genome assembly publicly available did not satisfy the initial minimal criteria of N50 (min 100kb, which is already low) that we have set for our analysis. Thus, we initially included *M. graminicola* and *M. chitwoodi* as diploid outgroups because highly contiguous genomes were available for these species.

However, we agree that *M. hapla* constitutes an even more interesting (and more closely related) outgroup and definitely makes the comparative analysis more complete.

Therefore, in this new version of the manuscript, we have made an exception to this 100kb N50 threshold to include *M. hapla* as well, owing to its high interest as a relatively close meiotic sexual and diploid outgroup.

We have included the genome (ca. 3400 scaffolds, N50 of 83.6kb) and predicted proteome (which is quite complete according to BUSCO / CEGMA) of *M. hapla* from the 2008 paper of our colleagues at NC State University in all our bioinformatics analyses.

This included re-doing the whole orthofinder analysis that now includes 69 nematode species.

We confirm in *M. hapla*:

- the absence of the canonical TTAGGC_n nematode telomeric repeats in the genome assembly
- the absence of telomerase and all the other telomere-associated genes that were also missing in the rest of the *Meloidogyne* species.
- the absence of the newly identified candidate telomeric repeats of *Minc*, *Mare*, and *Mjav* in the assembled *M. hapla* genome (and all the *Meloidogyne* genomes other than those of polyploid asexuals)

We have updated Figure 1, supplementary Figures 5 & 6 (which are now 6 & 7 in the revised supplement) as well as the results and methods accordingly.

We also completely agree that a long read genome (Nanopore or PacBio HiFi) for *M. hapla* is timely and would be useful for the whole community given the important biological, genetic, genomic and transcriptomic resources already available for this species.

We have been aware of multiple efforts including in the USA, in Europe and in China to sequence the *M. hapla* genome with long reads. This is why we have decided not to engage in a competition to publish a *M. hapla* genome starting from scratch while several projects are well advanced.

Instead, we have very recently been collaborating with UC Davis and UC Riverside (Shahid Siddique and Valerie Williamson) on the genome sequencing, assembly and analysis of *M. hapla* and as part of this collaboration, we are studying telomeres in this species as well. The results of this study will be part of another paper focused on the *M. hapla* genome and genetic map.

3. Why did the authors not use HiC analysis to bring the contig number even further down to chromosome number? Was that tried and it didn't work because of repeat structure? This is state-of-the-art in nematodes, and PacBio and HiC together result in chromosome number genomes throughout the phylum.

We agree that PacBio HiFi combined with HiC can result in chromosome-scale resolution genomes in several nematodes. We recently achieved this level of resolution in *Xiphinema index* which has a diploid genome and only 10 chromosomes and canonical telomeres, using PacBio HiFi + HiC, (paper in preparation). We also used PacBio HiFi in another population of *M. incognita* and in 10 different populations of *M. enterolobii*. In both cases, the genome assemblies yielded similar numbers of contigs and N50 values than for our ONT assemblies, suggesting polyploid genomes are challenging for genome assembly even with long reads. Furthermore, we realized that PacBio HiFi assemblies did not resolve the telomeric arrays as completely as the ONT assemblies did, resulting in many short contigs entirely composed of the repeats in the PacBio HiFi assemblies, probably as a result of reads being shorter than the whole length of the arrays. Therefore, in the present study we opted for ONT for the *Minc*, *Mare* and *Mjav* genomes.

As the reviewer suggested we were hopeful that HiC data would help scaffold the genome and approach chromosome scale resolution in these species.

Therefore, we generated Hi-C data for the same populations of *Minc*, *Mare* and *Mjav* as used for ONT sequencing. We used the Arima 2-enzymes strategy for the three species, and for *Minc* we also generated Hi-C data using the OmniC strategy.

We used multiple software (including SALSA2, YASH, 3D-DNA and Juicer) to try to scaffold the genome with the generated Hi-C data. Despite all our efforts, we realized that, indeed, and as guessed by Reviewer 3, the polyploid genome structure of *Minc*, *Mjav* and *Mare* with various degrees of sequence identity between the subgenomes poses major challenges to current scaffolding pipelines and software. In the three genomes, we observed blurry contact signals between the copies and several instances of fusions between copies resulting in scaffolds much bigger than the expected size of the biggest chromosomes. These fusions are likely due to false positive contact signal between the most similar subgenome copies and cast doubts on the validity of the other scaffolds.

Since the discovery of the new telomeric repeats, confirmed by FISH, was made possible by the assembly at the contig level and the scaffolds cannot be unambiguously defined at the moment, we have preferred to include the assembly at the contig level in this study. We will continue our efforts in scaffolding the genome by trying to use other strategies as well, or even develop new methods more adapted to polyploid genomes with variable degrees of identities between subgenomes. We might also try optical mapping although we are not convinced this will allow resolving ambiguities in very similar regions as this will probably result in the same optical barcodes.

4. The finding of the presumptive terminal repeats in around 50-60 contigs coincides with chromosome counts and the FISH data are very interesting. However, finding these repeats on only one end of the chromosomes, results in important questions. First, can the authors rule out the idea that Meloidogyne has circular chromosomes? That would also explain why you do not need telomerases etc. Do you see your repeats in the middle of some of your long reads? If so, that would be some indication for this unconventional idea.

Indeed, the terminal repeats we have identified are mostly at only one side of contigs and this has been validated by FISH on the chromosomes. This raises new questions about the structure of *Meloidogyne* chromosomes and whether they might be circular is an interesting hypothesis that could indeed explain the absence of telomerase and other features.

Circular chromosomes have never been formally described in nematodes but their existence cannot be excluded. In that context, *Diploscapter pachys* is an interesting case. Indeed, this species has only one pair of chromosomes, no canonical “TTAGGC” telomeric repeats, no evidence for a telomerase and is unable to do meiosis I division (Fradin H & al., Genome Architecture and Evolution of a Unichromosomal Asexual Nematode. *Curr Biol.* 2017.doi: 10.1016/j.cub.2017.08.038). Although in its condensed state, the chromosome is rather rod-shaped, the authors of the study conclude that “an interesting possibility that we cannot exclude is that the *D. pachys* chromosome is circular”. *M. incognita* is also unable to do ‘normal’ meiosis and has no simple telomeric repeat and telomerase. Although FISH results do not provide evidence for circular DNA when we observe chromosomes, this also cannot be completely excluded due to their tiny nature. Therefore, we conducted a comprehensive bioinformatics analysis in two steps to assess whether some elements could support the existence of circular chromosomes in this species.

(i) Firstly, we assessed whether split long-reads could support the existence of circular DNA as described in ECCsplorer (Mann L, & al., ECCsplorer: a pipeline to detect extrachromosomal circular DNA (eccDNA) from next-generation sequencing data. *BMC Bioinformatics.* 2022 Jan 14;23(1):40. doi: 10.1186/s12859-021-04545-2). We mapped the ONT long reads back to the Minc genome allowing for secondary alignments of reads spanning at least 1kb on the assembled contigs. Then, we identified 38,556 long reads that contained the complete telomeric motif 2-3-1 (coverage $\geq 90\%$) repeated at least two times. These reads ranged in size from 2-20kb with an average size of 15Kb. Cross-referencing the list of repeat-containing long reads with the mapping results yielded no identification of split mapping with one part of the read mapping on the contig region containing repeat arrays and another part of the read mapping elsewhere and at least 1 kb away from the repeat array region on the same contig (Additional Figure 6 below).

Figure 6; Illustration of the cases we were looking for to support the existence of circular chromosomes. No long read containing the telomeric repeats produced alignment on the genome assembly that would support circular DNA.

Hence, we found no evidence that contigs containing repeat arrays can represent circular chromosomes from read mapping results.

(ii) In a second step, we also investigated whether some reads had telomeric repeats in the middle as recommended by reviewer 3. From the pool of 38,556 ONT reads previously identified as containing a minimum of two complete motifs, we selected those that contained a repeat array flanked by at least 1kb of non-repetitive sequence upstream and downstream of the array. Applying this criterion, we identified only 163 such reads with putative central repeat. This represents only 0.005% of the reads and these reads predominantly exhibited a 1X coverage with a partial mapping while the average coverage in the genome is ca. 100X. This low frequency of reads with possibly internal repeats does not provide substantial support for the existence of circular chromosomes in *M. incognita*. Indeed, if chromosomes were mostly circular, we would expect a higher frequency of such reads and a coverage approaching the 100X average coverage for the rest of the genome.

Another possibility is that these few reads with possibly internal repeats could represent extrachromosomal circular DNA produced by telomeric regions. For instance, telomeric circles have been reported in the model nematode *C. elegans* through two-dimensional gel analysis and electron microscopy techniques (Tomaska L, et al., "Telomeric circles: universal players in telomere maintenance?" Nat Struct Mol Biol. 2009. doi: 10.1038/nsmb.1660). Whether this is the case in *M. incognita* as well would deserve further investigations in the future.

Overall, (i) the absence of split reads supporting circular contigs and (II) the extremely low frequency of reads with repeats in the middle provide no evidence for circular chromosomes in these *Meloidogyne* species.

5. Second, if the chromosomes would indeed not be circular, what are the sequences at the other end of the chromosomes? The authors could do easily a FISH analysis with sequences of their other contig ends. Given they have a round 300 contigs (minus the ones that carry the repeats, leaves 250 contigs). Just checking 10-20 contig sequences by FISH should yield some chromosome ends to address the question: Is the second chromosome end (if existing and not circular), identical/related between chromosomes, or different? Are there any motifs that have gone undetected so far?

Having ruled out the hypothesis of circular chromosomes, we agree that one important remaining question is what kind of sequence could be present at the other end.

In the initial version of the manuscript we performed the following analysis:

"First, we selected all contigs with a length >1Mb and already containing arrays of the composite repeat at one end. Using the same strategy as explained in methods, we searched for enriched motifs in the first or last 2kb at the opposite end. No evident motif could be identified as enriched, repeated and conserved at the other end of the bigger contigs. We extended this search to all contig beginnings or ends that were not already composed of the composite repeat but again, no evident enriched repeated motif could be found."

In addition to these initial analyses, we have now further refined and extended these searches by looking at contig extremities that seemed to display the same features as the telomeric repeat arrays already identified. According to Eugene gene predictions and EDTA repeat

predictions, we observed that all the regions containing arrays of our candidate telomeric repeats were rich in repeats but poor in genes and this is true for all the species (New Supplementary Table 6). Therefore we retrieved the other contig extremities that had similar repeat rich / gene poor ratio. We extracted 16kb, 10kb and 20kb respectively for *Minc*, *Mjav* and *Mare*, as this corresponded to the respective average telomeric repeat array sizes in the three species (now explained in Methods). Then, we ran a MEME search for enriched motifs using the same parameters as previously on these regions and verified the motifs found were specifically repeated at contig extremities. We also checked whether some repeats predicted by EDTA were specifically present in these regions.

All these searches (initial and new one on repeat rich / gene poor regions) resulted in identification of multiple different repeats enriched at contig extremities, none being present in a number of contig extremities close to the number of chromosomes. Instead, a multitude of repeats were each present on only a few contigs (usually 2 - 3 contigs with a maximum for a motif found in 6 contigs).

Now, it seems unlikely that other motifs have gone undetected and it does not seem that another evident repeat sequence is enriched at the other ends of most chromosomes.

Therefore, our hypothesis is that although one repeat is conserved at one extremity by almost all the chromosomes, different repeats are present at the other extremities and these repeats are not conserved between chromosomes but form protective heterochromatin at the other ends.

We have now proposed this hypothesis in the discussion.

These are all major concerns that the authors must address to present a complete story that would come close to a publication in a major journal. In the current form, this third generation *Meloidogyne* sequencing is just incomplete.

We thank reviewer 3 for all the points raised. We have taken into account all the comments and suggestions and we believe this has improved the overall quality and completeness of the manuscript.

REVIEWERS' COMMENTS

Reviewer #1 (Remarks to the Author):

This revised manuscript is much improved. I commend the authors for their extensive response to reviewer critiques and major revisions to the manuscript.

The authors have appropriately addressed my main concerns. The inclusion FISH and RNA-seq datasets for Mjav and Mare and the de novo strategy for repeat identification for each species separately help make the study less Minc-centric and reliant on that species as a starting point. I also appreciate the effort to evaluate the telomeric repeat evolution in the context of subgenome evolution. Furthermore, the inclusion of M. hapla at the suggestion of Reviewer 3 also strengthens the findings regarding repeats and loss of telomerase, and I understand why the authors did not include nanopore sequencing for this species.

I have only a few specific questions and minor comments.

In classifying the contigs as subgenome A vs. B, why was a phylogenetic approach not used? I don't think the Ks-based results are wrong (the subgenome differences are pretty apparent) but I wonder if there was an advantage over generating phylogenetic trees of genes coming from the triplet (or quadruplet) blocks.

Could the authors comment on why in Supp Fig 5a the Ks plot shapes look relatively similar across species? Wouldn't the two tetraploid species have 2 peaks of roughly equal high?
Why is M. luci is no longer in Figure 2?

Minor comments (line numbers in marked revision)

Line 592: The I in "In" is capitalized.

Line 684-688: This text could cite some references and perhaps define concerted evolution.

Line 761: "Reason" instead of "responsible"?

Line 1183: Typo "Supplemantary"

Figure S5 – panel d is mislabeled as c

The text in Supplemental Figure 8 for contig ends is "beg_" and "end_" while 9 & 10 are "Beg_" and "End_".

Reviewer #2 (Remarks to the Author):

The revised manuscript by Mota and colleagues has reasonably addressed many of the reviewers' concerns. The authors do a nice job of demonstrating novel telomere sequences and loss of telomerase in several polyploid Meloidogyne plant parasitic nematode species. This raises the exciting possibility that ALT telomere maintenance is active in these nematodes and might be a target for parasite control.

1. The authors present strong evidence that the unusual composite arrays of telomere sequences in Meloidogyne are coupled with loss of TERT and telomere binding proteins. When ALT evolves in cancer cells, telomere proteins remain because the canonical telomere sequence is present in ALT telomeres. Perhaps long periods of evolution in the context of ALT results in loss of telomere binding proteins or radical changes to their structures such that they can no longer be detected. Given the strong case for ALT, the abstract should probably mention lack of telomerase and the possibility that the novel telomere repeats reflect a form of ALT telomere maintenance.

2. The images shown in Fig 5 appear to be single planes within a nucleus. Could the authors please confirm that these are not maximal projections. I counted ~72 spots for Fig. 5a Mare, 69 spots for Mjav, 60 spots for Minc in prophase nuclei, based on the concept that both strong and weak telomere signals were likely to be distinct telomeres. These numbers are somewhat higher than reported by the authors in the last Results paragraph. It would be helpful if counts of telomere foci per nucleus were shown in Fig. 5, and if the authors could indicate in the results or methods that any telomere focus large or small was scored as a distinct telomere.

3. The authors state in the Discussion that 'a few chromosomes have repeats at both ends'. The authors show evidence that several chromosomes are likely to have telomeres at both ends. DNA sequencing for Minc that reveals 52 telomere repeat arrays for an estimated 46-47 chromosomes, which means that 5 or 6 chromosomes have unusual telomere repeats at both ends. If this is the case, then it is possible or even likely that all telomeres are the same, but that long read sequencing and microscopy have limitations in terms of defining telomeres in these species. While possible, it is difficult to envision a ALT mechanism that results in a specific set of telomere sequences at one chromosome end and not the other. Caveats that the authors might consider include:

During mitosis, telomeres might cluster or chromosomes might be pulled by their telomeres such that the two telomeres of each chromosome are adjacent to one another, and therefore might be physically indistinguishable by microscopy.

Perhaps telomeres and subtelomeric DNA of a number of chromosome termini are identical or almost identical, such that they cannot be distinguished on the basis of DNA sequence analysis. If this were the case, there would appear to be fewer telomeres than expected.

ALT telomere maintenance in cancer cells is meant to result in a broad range of telomere lengths, from very long to very short. Perhaps an explanation for the lack of apparent telomere sequence at some chromosome ends is that catastrophic events that result in loss of most or all of the telomere sequence occurs in Meloidogyne cells, and that telomere repeats are frequently added de novo to gene poor, repeat rich chromosome termini, whose subtelomeric sequences may consequently be unstable and evolve rapidly.

4. 'We never found another repeat unit present in a number of regions close to the number of chromosomes. Instead, we found multiple different repeats specific, each specific to a small number of contigs.' Is there an extra 'specific' after repeats? Please specify here that the different repeats were at the termini of a small number of contigs. Perhaps show what these repeats are and the number of contigs they cap in a supplemental table.

Reviewer #3 (Remarks to the Author):

The authors have addressed most of my concerns and suggestions quite well in their point-by-point response. Some of these answers have also been successfully integrated into the revised manuscript. However, I was disappointed to see that other important issues that would have made great contributions to the Discussion, have not been followed up. For example, I did not see any discussion on what the situation is at the second chromosome end that does not contain the repeat. Also at least mentioning the potential option of circular chromosomes in the discussion would have been good.

Reviewer #1 (Remarks to the Author):

This revised manuscript is much improved. I commend the authors for their extensive response to reviewer critiques and major revisions to the manuscript.

The authors have appropriately addressed my main concerns. The inclusion FISH and RNA-seq datasets for Mjav and Mare and the de novo strategy for repeat identification for each species separately help make the study less Minc-centric and reliant on that species as a starting point. I also appreciate the effort to evaluate the telomeric repeat evolution in the context of subgenome evolution. Furthermore, the inclusion of M. hapla at the suggestion of Reviewer 3 also strengthens the findings regarding repeats and loss of telomerase, and I understand why the authors did not include nanopore sequencing for this species.

We thank Reviewer 1 for the positive comments and the previous suggestions that helped improve the manuscript and make it more comprehensive.

I have only a few specific questions and minor comments.

In classifying the contigs as subgenome A vs. B, why was a phylogenetic approach not used? I don't think the Ks-based results are wrong (the subgenome differences are pretty apparent) but I wonder if there was an advantage over generating phylogenetic trees of genes coming from the triplet (or quadruplet) blocks.

Because we knew from Smudgeplot analyses that the Minc genome was AAB, this implied asymmetric relations with two more similar AA subgenomes and one more dissimilar B subgenome. We hypothesized that this asymmetric relation could be reflected by the average Ks value between triplets of blocks. This simple and straightforward calculation indeed allowed telling apart the different subgenomes. The same strategy was used to tell A and B subgenomes apart in Mjav and Mare with Ks used as a distance to produce classification trees.

Using phylogenomics analysis of concatenated alignments per sub-genome blocks is something we plan to do as soon as a long-read annotated genome is available for M. hapla. Indeed, as pointed out by reviewer 3, M. hapla has a particularly interesting phylogenetic position and represents a relatively close diploid meiotic outgroup. Actually, this is something we did in a previous analysis (Blanc-Mathieu et al. PLoS Genetics 2017) with more fragmented genomes. Our strategy will be to use M. hapla too to produce a more robust phylogenomics analysis as soon as an annotated long-read assembly is available. We agree with Reviewer 1 that this strategy would also allow defining which subgenomes are A and B and also reveal relations between Meloidogyne species.

Could the authors comment on why in Supp Fig 5a the Ks plot shapes look relatively similar across species? Wouldn't the two tetraploid species have 2 peaks of roughly equal high?

The reason for these similar 2-peak distributions is that whatever the genome, we always have twice as many higher Ks values than lower Ks values. Indeed, for Minc triplets, we have only one lower Ks relation (AA') and two higher Ks relations (AB and A'B). Similarly, for Mjav and Mare quadruplets, we systematically have two lower Ks relations (AA' and BB') and four higher Ks relations (AB, AB', A'B, A'B').

Why is M. luci is no longer in Figure 2?

As we only characterized *de novo* the motifs for the genomes we have assembled, we decided to not represent *M. luci* in this figure. Indeed, for *M. luci* we proceeded differently and searched for the presence of the motifs previously identified in Minc, Mare and Mjav. We found that the Minc motif was highly conserved in the Mluci genome and the results are presented separately in Supplementary Data 7.

Minor comments (line numbers in marked revision)

Line 592: The I in "In" is capitalized.

Corrected

Line 684-688: This text could cite some references and perhaps define concerted evolution.

We have added the following reference that defines and discusses concerted evolution in repetitive DNA "Elder Jr JF and Turner BJ (1995) Concerted evolution of repetitive DNA sequences in eukaryotes. Quarterly Review of Biology"

Line 761: "Reason" instead of "responsible"?

Corrected

Line 1183: Typo "Supplemantary"

Corrected

Figure S5 – panel d is mislabeled as c

Corrected

The text in Supplemental Figure 8 for contig ends is "beg_" and "end_" while 9 & 10 are "Beg_" and "End_".

Corrected

Reviewer #2 (Remarks to the Author):

The revised manuscript by Mota and colleagues has reasonably addressed many of the reviewers' concerns. The authors do a nice job of demonstrating novel telomere sequences and loss of telomerase in several polyploid Meloidogyne plant parasitic nematode species. This raises the exciting possibility that ALT telomere maintenance is active in these nematodes and might be a target for parasite control.

We thank Reviewer 2 for the positive comments and, indeed, this discovery of genus-specific unusual telomeric repeats suggesting ALT maintenance opens many perspectives, including for parasite control.

1. The authors present strong evidence that the unusual composite arrays of telomere sequences in Meloidogyne are coupled with loss of TERT and telomere binding proteins. When ALT evolves in cancer cells, telomere proteins remain because the canonical telomere sequence is present in ALT telomeres. Perhaps long periods of evolution in the context of ALT results in loss of telomere binding proteins or radical changes to their structures such that they can no longer be detected. Given the strong case for ALT, the abstract should probably mention lack of telomerase and the possibility that the novel telomere repeats reflect a form of ALT telomere maintenance.

We agree with reviewer 2 that this is an interesting point and in that perspective it is interesting to note that in *D. melanogaster* which also lacks a telomerase, the whole shelterin complex has been lost and replaced by a completely different set of proteins forming the Terminin complex. We think something similar has happened in these root-knot nematodes (and this is presented in the discussion). We have mentioned the absence of telomerase and orthologs of known telomeric DNA-binding proteins as well as possible alternative lengthening of telomeres (ALT) in the abstract.

2. The images shown in Fig 5 appear to be single planes within a nucleus. Could the authors please confirm that these are not maximal projections. I counted ~72 spots for Fig. 5a Mare, 69 spots for Mjav, 60 spots for Minc in prophase nuclei, based on the concept that both strong and weak telomere signals were likely to be distinct telomeres. These numbers are somewhat higher than reported by the authors in the last Results paragraph. It would be helpful if counts of telomere foci per nucleus were shown in Fig. 5, and if the authors could indicate in the results or methods that any telomere focus large or small was scored as a distinct telomere.

All confocal images (including Fig. 5) shown in the paper were processed in the same way and represent z-stack projection. For each nucleus, 6-8 z-stacks were created, and the maximum projection images were post-processed to achieve optimal intensity, which was applied to the

entire images. As there are large differences in signal intensity, this approach ensures that all signals are optimally visible. The quantification of the signals was performed in ImageJ. First, the color channels were separated and the default threshold was applied to the green channel. Then the spots were counted using the Particle Analysis tool for multiple nuclei and the average number is given in the Results section. There is a possibility that manual counting of only one displayed image results in slightly more spots, mainly due to the fact that some signals that overlap can be counted as separate signals and that very discrete signals are below the threshold to report distinct signals with sufficient confidence in the automatic counting method. However, we are convinced that analyzing more cyto-smears and reporting the average total number of spots provides the most reliable insight into the actual number of telomere signals. Additional details about signal counting can now also be found in the Methods section.

3. The authors state in the Discussion that 'a few chromosomes have repeats at both ends'. The authors show evidence that several chromosomes are likely to have telomeres at both ends. DNA sequencing for Minc that reveals 52 telomere repeat arrays for an estimated 46-47 chromosomes, which means that 5 or 6 chromosomes have unusual telomere repeats at both ends.

Indeed, 52 repeat arrays for 46-47 chromosomes in Minc would correspond to maximum 4 to 5 chromosomes with repeats at both ends, (and 42-43 at one single end).

If this is the case, then it is possible or even likely that all telomeres are the same, but that long read sequencing and microscopy have limitations in terms of defining telomeres in these species. While possible, it is difficult to envision a ALT mechanism that results in a specific set of telomere sequences at one chromosome end and not the other.

We have very consistent results between genome assemblies and FISH observations not only in Minc, but also in Mjav and Mare supporting the idea that most chromosomes have the repeat at one single end. Furthermore, a recent paper published in Nature Communications (Dai D. et al. 2023, <https://doi.org/10.1038/s41467-023-42700-w>, now cited in our manuscript) confirms these observations in Minc (Supplementary Figure 17). Therefore, although it is still possible, it seems quite unlikely to us that both the bioinformatics analysis of genome assemblies and the FISH observations in our study and in an independent study consistently missed repeats at both ends for most chromosomes.

We agree that it is surprising that one end of the chromosomes would be made of a conserved repeat and the other ends of a variety of different repeats. However, at this stage, this hypothesis is the one that best explains both the genome assemblies and DNA FISH observations.

Caveats that the authors might consider include:

During mitosis, telomeres might cluster or chromosomes might be pulled by their telomeres such that the two telomeres of each chromosome are adjacent to one another, and therefore might be physically indistinguishable by microscopy.

This is an interesting possibility, however, we never observed evidence for clustering of the telomeres forming a dense pack in the nuclei. We also do not observe chromosomes forming loops joined by their telomeres. Finally, observations in prophase also do not further support the repeat being mostly present at both ends. The number of detected fluorescence signals is close to the number of chromosomes for each species.

Perhaps telomeres and subtelomeric DNA of a number of chromosome termini are identical or almost identical, such that they cannot be distinguished on the basis of DNA sequence analysis. If this were the case, there would appear to be fewer telomeres than expected.

Although this cannot be completely excluded, this appears unlikely, considering that all our assemblies have been obtained on the longest nanopore reads of the highest quality for each genome. Even if telomeric and subtelomeric regions were highly identical and present at both ends, this would have resulted in telomeric repeats shorter than reality in the assemblies at both ends but not systematically missed at one end.

ALT telomere maintenance in cancer cells is meant to result in a broad range of telomere lengths, from very long to very short. Perhaps an explanation for the lack of apparent telomere sequence at some chromosome ends is that catastrophic events that result in loss of most or all of the telomere sequence occurs in *Meloidogyne* cells, and that telomere repeats are frequently added de novo to gene poor, repeat rich chromosome termini, whose subtelomeric sequences may consequently be unstable and evolve rapidly.

We thank reviewer 2 for this interesting remark and suggestion. This could indeed be an explanation for the repeated observations of one or two chromosomes without a fluorescence signal at all. The discovery of these unusual telomeric repeats in *Meloidogyne* species opens many perspectives and a lot remains to be described. The study of the dynamics of telomere length and variations between chromosomes / populations / life stages is part of the experiments we plan to conduct in the future. The mechanisms involved in the multiplication and stability of these repeats remain to be described as well.

4. 'We never found another repeat unit present in a number of regions close to the number of chromosomes. Instead, we found multiple different repeats specific, each specific to a small number of contigs.' Is there an extra 'specific' after repeats?

We thank reviewer 2 for having identified this error and we have corrected it.

Please specify here that the different repeats were at the termini of a small number of contigs.

Thanks for this suggestion, we have added this information to clarify the message.

Perhaps show what these repeats are and the number of contigs they cap in a supplemental table.

We think the other identified motifs would need further investigation and because they are not at the same level of study as the motifs presented in the manuscript, we prefer not to include them at the moment. For instance, some of these motifs are mostly present repeated on short contigs with some covering the whole contig length. Whether they represent unassembled putative telomeric repeat or extra-chromosomal repetitive regions remains to be clarified. For the most promising repeats, we will probably conduct DNA FISH analyses as well, in the future. As an example, a promising motif identified in *Minc* is indicated below. The repeat is found at the extremity of 7 contigs plus 3 fully repetitive contigs. Interestingly, two contigs contain the telomeric motif presented in the manuscript at the other end (contigs 20 and 47 in bold).

We will continue our investigations on the other end in the future.

Motif	Contig extremities
	Beg_20, Beg_33, End_44, Beg_47, Beg_93, Beg_160, Beg_End_228, Full_158, Full_206, Full_263

Reviewer #3 (Remarks to the Author):

The authors have addressed most of my concerns and suggestions quite well in their point-by-point response. Some of these answers have also been successfully integrated into the revised manuscript.

We thank reviewer 3 for the suggestions and comments which helped improve the manuscript.

However, I was disappointed to see that other important issues that would have made great contributions to the Discussion, have not been followed up. For example, I did not see any discussion on what the situation is at the second chromosome end that does not contain the repeat. Also at least mentioning the potential option of circular chromosomes in the discussion would have been good.

We completely agree that this would constitute an interesting contribution to the discussion and have now added these points to the manuscript discussion section.

"This ensemble of observations supports the repeats we identified are surprisingly at only one end of most chromosomes. Although we identified multiple other contig ends which are repeat-rich and gene-poor we did not find another evident repeat unit forming a number of arrays close to the number of chromosomes in any species. Therefore, we hypothesize that multiple different repeats, each being specific to a few chromosomes, are present at the other end and form protective heterochromatin. An alternative hypothesis would be that *Meloidogyne* chromosomes form loops or circular structures. This has been evoked for the uni-chromosomal nematode *D. pachys* which also lacks telomerase(11). However, no evidence from the genome reads or chromosomal observations supports this possibility at this stage. Further investigations will be necessary to characterize what kind of DNA or structure protects the other ends of most chromosomes."